# Persistent humoral immune response in youth throughout the COVID-19 pandemic: prospective school-based cohort study

Alessia Raineri[1], Thomas Radtke[1], Sonja Rueegg[1], Sarah R. Haile[1], Dominik Menges[1], Tala Ballouz[1], Agne Ulyte[1], Jan Fehr[1], Daniel L. Cornejo[1], Giuseppe Pantaleo[2], Céline Pellaton[2], Craig Fenwick[2], Milo A. Puhan[1] & Susi Kriemler[1] ✉

Understanding the development of humoral immune responses of children and adolescents to SARS-CoV-2 is essential for designing effective public health measures. Here we examine the changes of humoral immune response in school-aged children and adolescents during the COVID-19 pandemic (June 2020 to July 2022), with a specific interest in the Omicron variant (beginning of 2022). In our study "Ciao Corona", we assess in each of the five testing rounds between 1874 and 2500 children and adolescents from 55 schools in the canton of Zurich with a particular focus on a longitudinal cohort (n=751). By July 2022, 96.9% (95% credible interval 95.3–98.1%) of children and adolescents have SARS-CoV-2 anti-spike IgG (S-IgG) antibodies. Those with hybrid immunity or vaccination have higher S-IgG titres and stronger neutralising responses against Wildtype, Delta and Omicron BA.1 variants compared to those infected but unvaccinated. S-IgG persist over 18 months in 93% of children and adolescents. During the study period one adolescent was hospitalised for less than 24 hours possibly related to an acute SARS-CoV-2 infection. These findings show that the Omicron wave and the rollout of vaccines boosted S-IgG titres and neutralising capacity. Trial registration number: NCT04448717. https://clinicaltrials.gov/ct2/show/NCT04448717.

Monitoring the evolution of seroprevalence and assessing changes in humoral immune responses against severe acute respiratory syndrome coronavirus type 2 (SARS-CoV-2) in children and adolescents over time is important to understand the evolution of the pandemic and to inform public health measures, including vaccination strategies and preventive measures at school.

Several studies were conducted to detect SARS-CoV-2 infections in children and adolescents and to determine seroprevalence at different times of the pandemic[1–9]. However, little is known about the development and persistence of humoral immune responses over time as most studies were cross-sectional[1–5]. A systematic review[10]

reported the persistence of cellular and humoral immune responses in children and adolescents during the pre-Omicron period, lasting for at least 10 to 12 months. Meanwhile, few studies[7,11] focused on immune responses following infection with the Omicron variant, addressing neutralising activity and differentiating between infection, vaccination, or both. These studies showed that the combination of SARS-CoV-2 infection and vaccination showed the highest immune responses in children and adolescents compared to those infected but unvaccinated. One of the studies[7] found that nearly all children and adolescents between 6 and 17 years of age had anti-spike IgG antibodies, but neutralising capacity against Omicron

[1]Epidemiology, Biostatistics and Prevention Institute (EBPI), University of Zurich, Hirschengraben 84, 8001 Zürich, Zurich, Switzerland. [2]Service of Immunology and Allergy, Lausanne University Hospital (CHUV), University of Lausanne (UNIL), Lausanne, Switzerland. ✉e-mail: susi.kriemlerwiget@uzh.ch

was much lower in children (<12 years) compared to adolescents (≥12 years).

Many countries started to administer COVID-19 vaccines to children and adolescents in 2021 to 2022, after trials demonstrated the effectiveness of the COVID-19 vaccine against reinfection[12,13] and severe disease[14], and vaccines were approved for use in these populations by the Food and Drug Administration and European Medicines Agency. In Switzerland, the COVID-19 vaccine was available for adolescents aged 12 years and older by mid-2021 and for children aged 5 to 11 years in early 2022[15].

In early 2022, the high incidence of SARS-CoV-2 infections in children and adolescents due to the Omicron variant raised concerns, as infections spread despite the rollout of vaccines in that population[16–18]. In that period, the coincidence of incomplete immunisation of children and adolescents with the highly transmissible Omicron variant strongly determined the further evolution of the seroprevalence as well as the longitudinal development of humoral immune responses[18].

In this observational school-based study, we aimed to assess the longitudinal development of the humoral immune response against SARS-CoV-2 in school-aged children and adolescents throughout the COVID-19 pandemic from June 2020 to July 2022. In particular, we focused on how anti-spike IgG antibodies and neutralising response changed during the first peak of infections with the Omicron variant in the context of (re-)infections, vaccinations, and their combination.

## Results

### Participant characteristics

Over the course of the study, we tested between 1874 and 2500 children and adolescents over five testing rounds between June 2020 and July 2022 (flowchart in Supplementary Fig. 1). The participation rate within classes ranged between 36% and 50% across all testing rounds (Supplementary Table 1).

Seroprevalence in children and adolescents increased with each testing round (Supplementary Fig. 2). The largest increase in

seroprevalence occured between T4 (Nov/Dec 2021) and T5 (Jun/Jul 2022) from 46.5% [95% credible interval [CrI] 42.5–51.3%] to 96.9% [95% CrI 95.3–98.1%] in the overall study population and from 31.3% [95% CrI 27.5–35.9%] to 95.7% [95% CrI 93.0–97.7%] among unvaccinated children and adolescents (Supplementary Table 2).

During the entire study period, a total of three seropositive children and adolescents reported hospital stays, all of which lasted less than 24 h and of which one was possibly related to an acute SARS-CoV-2 infection (Supplementary Table 3 for details).

The longitudinal cohort comprised 751 children and adolescents who participated in the last testing round (T5 Jun/Jul 2022) as well as in at least three previous testing rounds (Table 1, for details on chronic conditions, see Supplementary Table 4). We categorised children and adolescents based on their exposure status, which was defined as follows: children and adolescents never testing positive for anti-spike IgG were categorised as negative, unvaccinated children and adolescents ever testing positive for anti-spike IgG as infected, vaccinated children and adolescents testing always negative for anti-spike IgG prior to vaccination and never testing positive for anti-nucleocapsid IgG as vaccinated, and children and adolescents testing seropositive before getting vaccinated, or were vaccinated and tested positive for anti-nucleocapsid-IgG antibodies as hybrid.

### Trajectory of anti-spike IgG antibodies

Figure 1 shows the trajectory of anti-spike IgG antibodies in the longitudinal cohort ($n = 386$ participants), without (Fig. 1a) and with (Fig. 1b) categorisation based on exposure status (i.e., negative, infected, vaccinated, hybrid). We excluded children and adolescents who never tested seropositive throughout all five testing rounds ($n = 37$ participants) and children and adolescents who seroconverted between T4 (Nov/Dec 2021) and T5 (Jun/Jul 2022) ($n = 328$ participants). Participants were categorised according to their time of seroconversion, i.e., group 1 seroconverted before T1 (Jun/Jul 2020), group 2 seroconverted between T1 (Jun/Jul 2020) and T2 (Oct/Nov 2020), group 3 seroconverted between T2 (Oct/Nov 2020) and T3

**Table 1 | Baseline characteristics of the longitudinal study population at each testing round**

| | T1 | T2 | T3 | T4 | T5 |
|---|---|---|---|---|---|
| Timeframe of testing | Jun/Jul 2020 | Oct/Nov 2020 | Mar/Apr 2021 | Nov/Dec 2021 | Jun/Jul 2022 |
| Predominant VOC[a] | Wildtype | Wildtype | Alpha | Delta | Omicron |
| N unique[b] | 751 | | | | |
| N tested[c] | 695 | 725 | 738 | 722 | 751 |
| Age[f] (years) | 10 (8–13) | 11 (9–13) | 11 (9–24) | 12 (10–15) | 12 (10–15) |
| Sex (n, % male) | 326 (47%) | 341 (47%) | 352 (48%) | 341 (47%) | 355 (47%) |
| Age group | | | | | |
| <12 years | 496 (71%) | 492 (68%) | 465 (63%) | 377 (52%) | 325 (43%) |
| ≥12 years | 199 (29%) | 233 (32%) | 273 (37%) | 345 (48%) | 426 (56%) |
| Chronic conditions[d] | 150 (21%) | 161 (22%) | 163 (22%) | 159 (22%) | 166 (23%) |
| Vaccinated[e] | | | | | |
| Overall | 0 | 0 | 0 | 184/722 | 345/751 |
| <12 years | | | | (25%) | (46%) |
| ≥12 years | | | | 0 | 93/325 (29%) |
| | | | | 184/345 | 252/426 |
| | | | | (53%) | (59%) |
| Questionnaires completed | 656 (94%) | 634 (87%) | 626 (85%) | 595 (82%) | 545 (73%) |

[a]Predominant variant of concern (VOC) in Switzerland (>50% of circulating variants in Switzerland).
[b]Unique number of children and adolescents tested throughout the entire study period.
[c]Number of children and adolescents tested per round.
[d]Details on chronic conditions can be found in Supplementary Table 4.
[e]Grouped into <12 years and ≥12 years, since in Switzerland, adolescents ≥12 years of age could get vaccinated since mid-June 2021 and children between 5 and 11 years of age from January 2022. Source data are provided as a Source Data file.
[f]Median (interquartile range); T1: Jun/Jul 2020; T2: Oct/Nov 2020; T3: Mar/Apr 2021; T4: Nov/Dec 2021; T5: Jun/Jul 2022.

## a

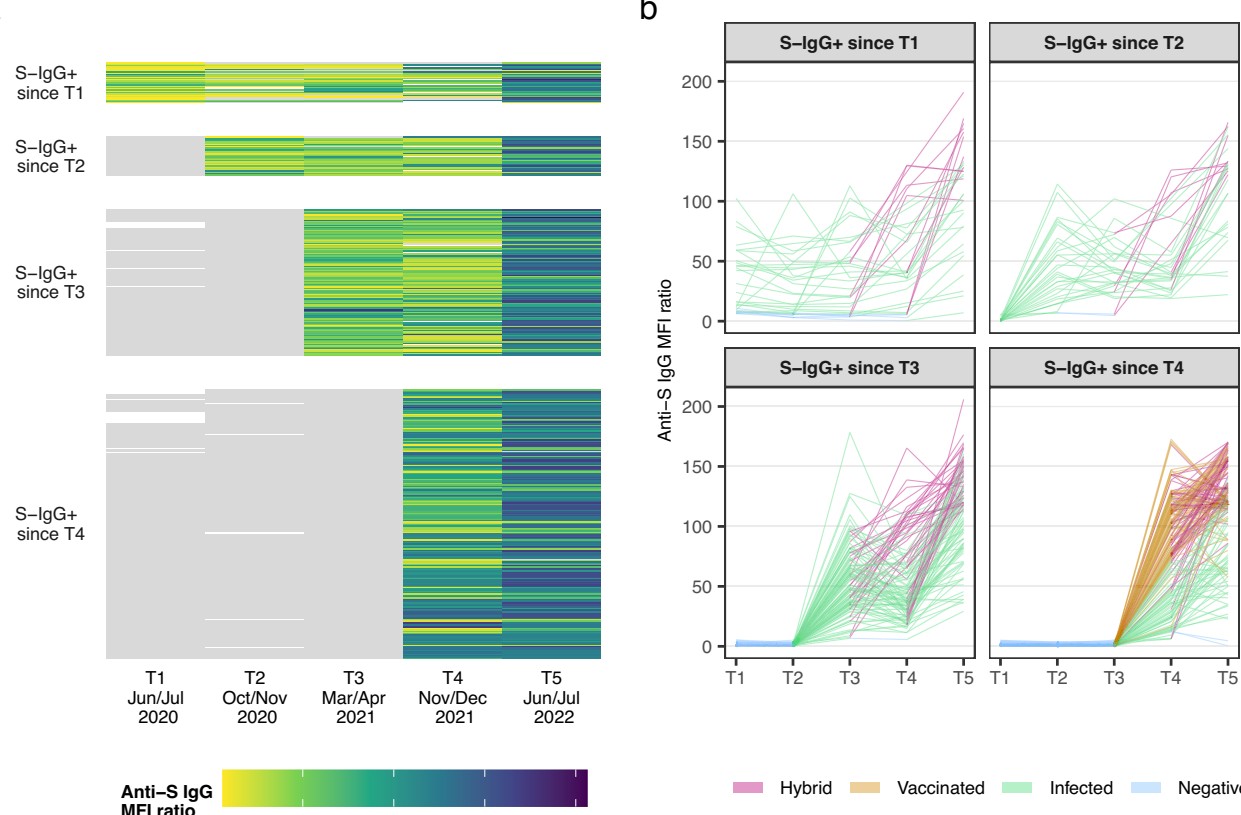

## b

**Fig. 1 | Individual trajectories of anti-spike IgG (S-IgG) mean fluorescence intensity (MFI) ratios over time separated by the first incidence of seropositive result (*n* = 386 participants).** S-IgG+: Anti-spike IgG positive mean fluorescence intensity (MFI) ratio; (1) S-IgG+ since T1: *n* = 32 participants; (2) S-IgG+ since T2: *n* = 30 participants; (3) S-IgG+ since T3: *n* = 114 participants; (4) S-IgG+ since T4: *n* = 210 participants. Children and adolescents seroconverting from T4 to T5 are not shown (*n* = 328 participants). **a** The heatmap shows the changes in titres through colour changes. Grey denotes seronegative anti-spike IgG result (MFI ratio <6). Colour denotes seropositive anti-spike IgG result with different MFI titre levels. The white colour indicates no serology result available. 37 children and adolescents tested seronegative throughout all five testing rounds are not shown in the figure.

**b** This figure shows the individual trajectories of anti-spike IgG titres, coloured by a child or adolescents' exposure status over time (i.e., hybrid (violet), vaccinated (orange), infected (green), negative (blue)). Negative denotes testing negative for anti-spike IgG; infected denotes unvaccinated individuals testing positive for anti-spike IgG; vaccinated denotes vaccinated individuals testing always negative for anti-spike IgG prior to vaccination and not testing positive for anti-nucleocapsid IgG; hybrid denotes individuals testing seropositive before getting vaccinated or were vaccinated and tested positive for anti-nucleocapsid-IgG antibodies. Source data are provided as a Source Data file.T1: Jun/Jul 2020; T2: Oct/Nov 2020; T3: Mar/Apr 2021; T4: Nov/Dec 2021; T5: Jun/Jul 2022.

(Mar/Apr 2021), and group 4 seroconverted between T3 (Mar/Apr 2021) and T4 (Nov/Dec 2021). After seroconversion, anti-spike IgG antibodies remained detectable at 6 months (T4 to T5) in: 99.0% [confidence interval [CI] 96.6 to 99.7%; *n* = 208/210 participants], at 12 months (T3 to T5) in: 99.1% [95% CI 95.2 to 99.8%; *n* = 113/114 participants], at 18 months (T2 to T5) in: 93.3% [95% CI 78.7 to 98.2%; *n* = 28/30 participants] and at 24 months (T1 to T5) in: 68.8% [95% CI 51.4 to 82.0%; *n* = 22/32 participants] in children and adolescents (Fig. 1a). At T5, antibodies were detectable in 99.5% (*n* = 384/386 participants) of children and adolescents who seroconverted in any previous testing round (Supplementary Fig. 3 shows the mean fluorescence intensity (MFI) ratio converted to WHO units per millilitre (U/ml) scale as measured by the Elecsys Anti-SARS-CoV-2 immunoassay by Roche). Anti-spike IgG titres increased with each testing round either by (re-)infection, vaccination, or a combination of the two (Fig. 1). A first increase in antibody titres occurred between T3 (median MFI ratio of 48.5 [interquartile range (IQR): 33.6 to 68.8]) and T4 (MFI ratio of 74.0 [IQR: 36.5 to 109.0]), coinciding with the introduction of vaccination in the 12 years and older age group in Switzerland. The highest increase in titres, visualised by the most substantial colour change in Fig. 1, occurred between T4 (MFI ratio of 74.0 [IQR: 36.5 to 109.0]) and T5 (MFI ratio of 122.0 [IQR: 95.2 to 144.0])

when Omicron became the predominant variant of concern (VOC) in Switzerland.

We were also interested in estimating the decay of infection-elicited anti-spike IgG antibodies. For these analyses, we only included infected (based on exposure status) children and adolescents. We excluded all children and adolescents with potential reinfection, defined here as the presence of a newly positive anti-nucleocapsid IgG antibody or any increase in anti-spike IgG titres between two testing points. We then estimated the anti-spike IgG antibody half-life, over two time frames for 365 and 220 days. The longer time frame (365 days) was chosen as it was the longest possible time frame based on the study duration and the shorter time frame was selected for comparability with the published literature. The anti-spike IgG half-life estimate for the longer time frame was 305 days [95% CI 263–363 days] (Supplementary Fig. 4a) and for the shorter time frame 220 days [95% CI 170–312 days] (Supplementary Fig. 4b).

Given our primary interest in the Omicron wave, the main analysis included the longitudinal cohort consisting of children and adolescents that participated in the last (T5) and in at least three further testing rounds. Additionally, we performed a sensitivity analysis including all children and adolescents with data for at least four testing rounds (regardless of participation in T5). The half-life estimates in the

sensitivity analysis were comparable to those in the main analysis for the longer time frame of 365 days (284 days [95% CI 245–337 days] vs. 305 days [95% CI 263–363 days]) and for the shorter time frame of 220 days (204 days [157–290 days] vs. 220 days [95% CI 170–312 days], Supplementary Fig. 4c, d). Children and adolescents for the main and the sensitivity analyses including the longer and shorter time frame were comparable among cohorts for sex distribution (main analysis: for the longer time frame 51 (45%) were male vs. shorter time frame 22 (42%) were male; sensitivity analysis: for the longer time frame 59 (45%) were male vs. shorter time frame 26 (43%) were male) and age (main analysis: for the longer time frame 11 years [IQR: 9–12 years] vs. shorter time frame 11 years [IQR: 9–13 years]; sensitivity analysis: for the longer time frame 11 years [IQR: 9–12 years] vs. shorter time frame 11 years [IQR: 9–14 years]).

### Effect of the Omicron wave: evolution of anti-spike IgG antibodies

Figure 2 shows the evolution of anti-spike IgG antibodies of children and adolescents separated by their serology and exposure status (i.e., negative, infected, vaccinated, hybrid) at T4 (Nov/Dec 2021) and followed by T5 (Jun/Jul 2022). These testing rounds corresponded to the time when Omicron started to be the dominant VOC in Switzerland until the end of the first peak in incidence (Supplementary Fig. 2). Children and adolescents with hybrid immunity and those who were vaccinated showed similarly high titres at T4 or T5, whereas those infected showed considerably lower titres (Supplementary Tables 5 or 6). On average titres of all subgroups increased from T4 to T5. The highest increase in titres was seen in children and adolescents who were infected or negative in T4 and received their first vaccination between T4 and T5. The smallest increase and the lowest titres in T4 and T5 were observed in children and adolescents who were negative at T4 and were infected with SARS-CoV-2 for the first time between T4 and T5 (Supplementary Fig. 5 and Supplementary Table 6a, b with original MFI ratios as well as MFI ratios converted to WHO U/ml). Among vaccinated children and adolescents, those with a vaccination only prior to T4, showed similar titres at T5 compared to those additionally vaccinated between T4 and T5 (median MFI ratio of 125.1 [IQR: 117.4-151.6] vs. 128.5 [IQR: 119-156.1], respectively). We found no differences in titres when stratifying children and adolescents by age (<12 and ≥12 years) (Supplementary Fig. 6a–d and Supplementary Table 7a–d).

To quantify the proportions of children and adolescents with prior infection, vaccination, or both at T4 who experienced an infection or reinfection between T4 and T5, we used changes in antibody titres to determine (re-)infection events due to the high number of undiagnosed infections during the Omicron wave. We only included children and adolescents who were seropositive due to infection, vaccination, or both at T4. We divided those children and adolescents into infected or vaccinated, irrespective of infection prior to T4. We further divided the vaccinated into those with older vaccination (last vaccination prior to T4) and recent vaccination (last vaccination between T4 and T5). To detect (re-)infections, we tested for a ≥25% increase in anti-spike IgG titres and the presence of anti-nucleocapsid IgG antibodies between T4 and T5. The latter are expressed only after a SARS-CoV-2 infection, but not after a vaccination. The 25% threshold determined prior was judged as a relevant increase consistent with a (re-)infection (in the absence of vaccination) between two testing points using data from multiple population-based cohorts throughout the pandemic and considering any potential measurement errors. For infected children and adolescents and those with older vaccination (last vaccine before T4), we considered a 25% or higher increase in titres and/or the existence of anti-nucleocapsid IgG antibodies. Whereas for those with recent vaccination (vaccination between T4 and T5), only the existence of anti-nucleocapsid IgG was considered since the increase in anti-spike IgG titres could be attributed to the

recent vaccination (Supplementary Table 8a). By mid-2022, reinfections were observed in 90.2% of infected children and adolescents (n = 314 participants), 75.2% in those with older vaccination (n = 206 participants) and in 36.4% among those with recent vaccination (n = 221 participants, Supplementary Table 8a). To test for the robustness of the ≥25% increase, we performed sensitivity analyses with two alternative thresholds (15% and 35%) and found that percentages of (re-)infections did not differ significantly from the 25% threshold (Supplementary Table 8b).

### Effect of the Omicron wave: evolution of the neutralising antibodies

Figure 3 shows the development of neutralising antibodies against different SARS-CoV-2 variants (Wildtype, Delta, and Omicron BA.1) between T4 (Nov/Dec 2021) and T5 (Jun/Jul 2022). We again examined participants by their serology at T4 and exposure status (i.e., negative, infected, vaccinated, hybrid). In general, neutralising activity increased between T4 and T5, regardless of exposure status. The neutralising response was proportionally higher, but comparable in those with hybrid immunity and vaccination only (e.g., anti-Omicron BA.1 at T5 98.9% [95% CI 96.0-99.7%] and 81.6% [95% CI 74.9–86.9%], respectively, Supplementary Table 5), but lower in infected participants (e.g., anti-Omicron BA.1 at T5 64.9% [95% CI 59.8–69.7%]). Children and adolescents infected at T4 (Fig. 3i) compared to those negative at T4 (Fig. 3l) showed higher neutralising response at T5 (Supplementary Table 9 with detailed test results). Among vaccinated children and adolescents, those with a vaccination only prior to T4, showed proportionally similar neutralising activity at T5 compared to those additionally vaccinated between T4 and T5 (e.g., anti-Omicron BA.1 93.6% [95% CI 82.8–97.8%] vs. 73.0% [95% CI 57.0–84.6%], respectively). We found no differences in neutralising responses when stratifying children and adolescents by age (<12 and ≥12 years) (Supplementary Fig. 7 and Supplementary Table 10).

The neutralising response was highest against Wildtype SARS-CoV-2, followed by the Delta and Omicron BA.1 variant, except in children and adolescents getting newly infected between T4 to T5 (Fig. 3l), where the neutralising response was highest against Omicron BA.1, followed by responses against Wildtype and Delta.

## Discussion

Throughout the course of the Ciao Corona study lasting from June 2020 to July 2022, seroprevalence increased with the spread of the Delta and Omicron BA.1 variant and the uptake of vaccination in children and adolescents. By July 2022 (after the first peak of the Omicron wave) despite limited uptake of vaccination (58% in those ≥12 years, 28% in <12 years), 96.9% [95% CrI 95.3–98.1%] of all children and adolescents had anti-spike IgG antibodies against SARS-CoV-2 and most children under the age of 12 became seropositive. 93% of children and adolescents who seroconverted early in the pandemic were persistently seropositive for up to 18 months. Vaccinated children and adolescents—regardless of prior infection—had high to very high anti-spike IgG titres and proportionally higher neutralising response, compared to unvaccinated but infected children and adolescents. These findings were independent of the timing of vaccinations and age groups. Reinfections occurred more frequently in infected (i.e., vaccine-naïve) than in vaccinated children and adolescents. Throughout the study, only one adolescent was hospitalised for less than 24 h due to a SARS-CoV-2 infection.

Several studies reported on the persistence of antibodies after a SARS-CoV-2 infection in children and adolescents with some reporting detectable anti-spike IgG levels over 4 to 6 months[19–22], or even up to 9 to 18 months[23–26]. In our study, children and adolescents who seroconverted early in the COVID-19 pandemic were persistently seropositive for up to 18 months. We estimated that the anti-spike IgG half-life was 305 days in infected (i.e., vaccine-naïve) children and

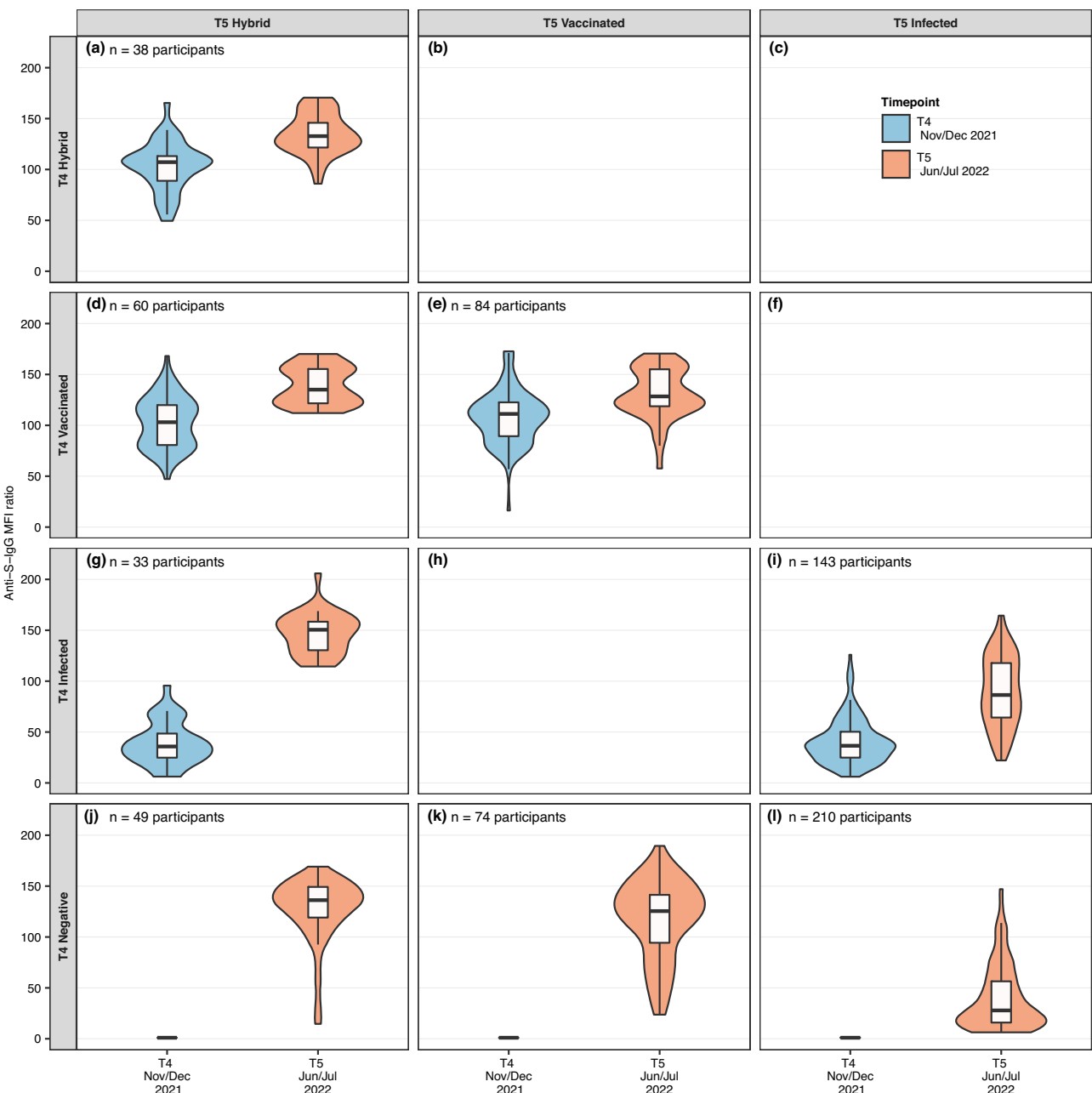

**Fig. 2 | Evolution of anti-spike IgG mean fluorescence intensity (MFI) ratios between T4 (Nov/Dec 2021) and T5 (June/July 2022) among negative, infected, vaccinated children and adolescents, and those with hybrid immunity.** The evolution of anti-spike IgG mean fluorescence intensity (MFI) ratios is shown as violin plots display mirrored density for each titre value (continuous distribution). Children and adolescents are categorised based on their exposure status (i.e., hybrid, vaccinated, infected, negative). Panel **a** 38 participants with hybrid immunity at T4 and T5, who had a median MFI titre of 107.1 in T4 (blue) and increased to a median MFI titre of 133.1 in T5 (red). Panel **b** No children were observed in this panel, as it is highly unlikely to be hybrid in T4 and later only show evidence of vaccination at T5. Panel **c** No children were observed in this panel, as it is highly unlikely to be hybrid in T4 and later only show evidence of infection at T5. Panel **d** 60 participants vaccinated at T4 and with hybrid immunity at T5. Panel **e** 84 participants vaccinated at T4 and T5. Panel **f** No children were observed in this panel, as it is highly unlikely to be vaccinated in T4 and later only show evidence of infection at T5. Panel **g** 33 participants infected at T4 and with hybrid immunity at

T5. Panel **h** No children were observed in this panel. Panel **i** 143 participants infected at T4 and T5. Panel **j** 49 participants negative at T4 and with hybrid immunity at T5. Panel **k** 74 participants were negative at T4 and vaccinated at T5. Panel **l** 210 participants negative at T4 and infected at T5. Negative denotes seronegative at T4; Infected denotes seropositive and unvaccinated (T4 infected denotes seropositive based on anti-spike IgG prior to the T4 testing and unvaccinated, T5 infected denotes seropositive based on anti-spike IgG between T4 and T5 and unvaccinated); Vaccinated denotes vaccinated participants but negative in previous rounds and without evidence for anti-nucleocapsid IgG response. Participants with hybrid immunity were seropositive before getting vaccinated or were vaccinated and tested positive for anti-nucleocapsid-IgG antibodies. Titre levels at T4 and at T5 are shown in blue and orange, respectively. Boxplots in panels show the median and interquartile range (IQR; whisker: 1.5 IQR). 60 children and adolescents are not shown in the figure ($n$ = 31 participants were seronegative at T4 and T5, $n$ = 29 participants had no data at T4). Source data are provided as a Source Data file.

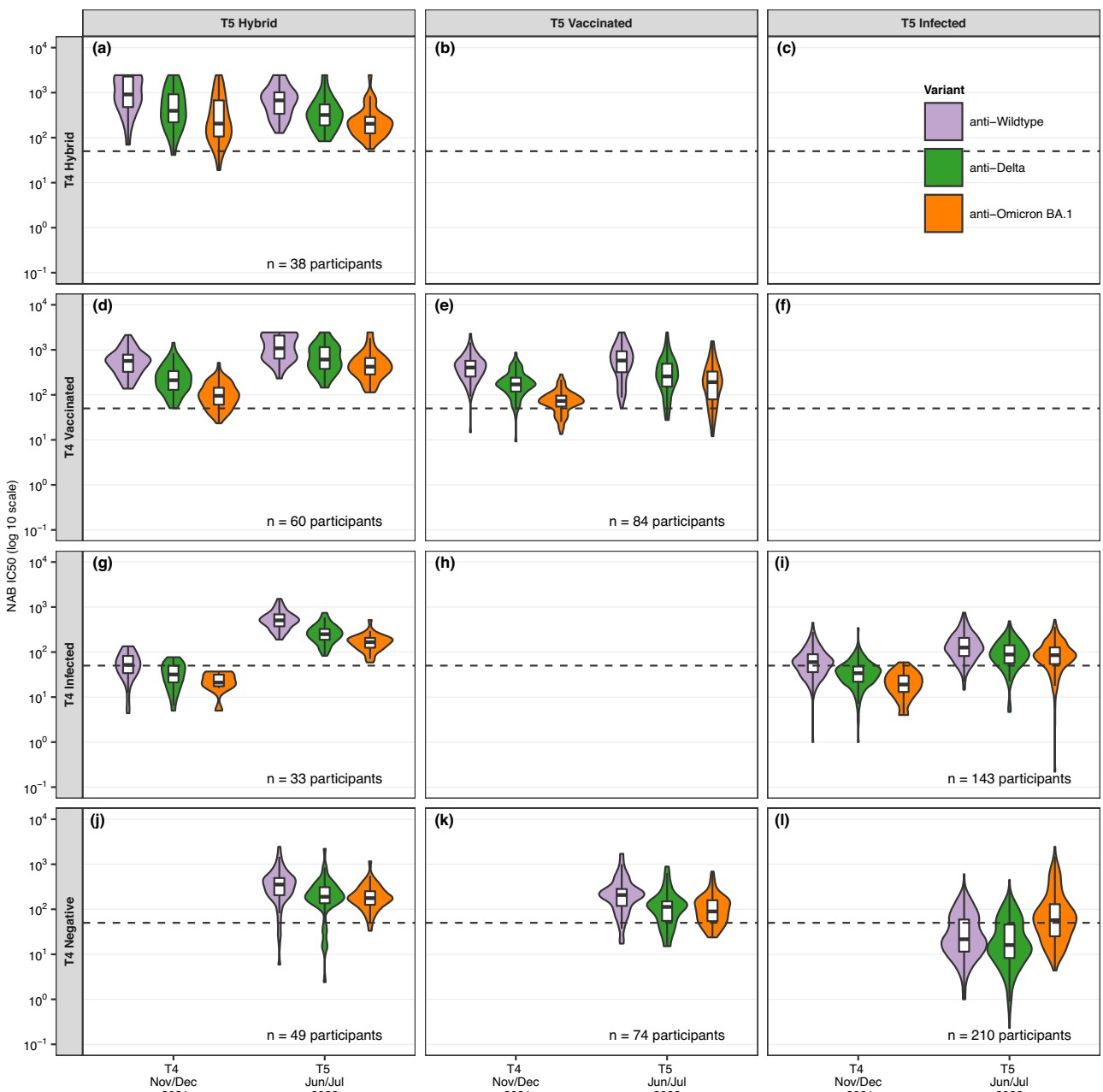

**Fig. 3 | Evolution of neutralising antibody (NAB) half maximal inhibitory concentrations (IC50) between T4 (Nov/Dec 2021) and T5 (June/July 2022) among negative, infected, vaccinated children and adolescents, and those with hybrid immunity.** The evolution of neutralising antibodies (NAB) is shown by violin plots that display mirrored density for each NAB half maximal inhibitory concentration (IC50) (continuous distribution). Children and adolescents are categorised based on their exposure status (i.e., hybrid, vaccinated, infected, negative). Panel **a** 38 participants with hybrid immunity at T4 and T5. Panel **b** No children were observed in this panel, as it is highly unlikely to be hybrid in T4 and later only show evidence of vaccination at T5. Panel **c** No children were observed in this panel, as it is highly unlikely to be hybrid in T4 and later only show evidence of infection at T5. Panel **d** 60 participants vaccinated at T4 and with hybrid immunity at T5. Panel **e** 84 participants vaccinated at T4 and T5. Panel **f** No children were observed in this panel, as it is highly unlikely to be vaccinated in T4 and later only show evidence of infection at T5. Panel **g** 33 participants infected at T4 and with hybrid immunity at T5. Panel **h** No children were observed in this panel. Panel **i** 143 participants infected

at T4 and T5. Panel **j** 49 participants negative at T4 and with hybrid immunity at T5. Panel **k** 74 participants were negative at T4 and vaccinated at T5. Panel **l** 210 participants negative at T4 and infected at T5. Negative denotes seronegative at T4; Infected denotes seropositive and unvaccinated (T4 infected denotes seropositive based on anti-spike IgG prior to the T4 testing and unvaccinated, T5 infected denotes seropositive based on anti-spike IgG between T4 and T5 and unvaccinated); Vaccinated denotes vaccinated participants but negative in previous rounds and without evidence for anti-nucleocapsid IgG response. Children and adolescents with hybrid immunity were seropositive before getting vaccinated or were vaccinated and tested positive for anti-nucleocapsid-IgG antibodies. The dotted line indicates the NAB IC50 value threshold (50) for neutralising activity. Children and adolescents with NAB IC50 values above the threshold are assumed to have 50% or higher neutralisation capacity. Boxplots in panels show the median and interquartile range (IQR; whisker: 1.5 IQR). 60 children and adolescents are not shown in the figure ($n = 31$ participants were seronegative at T4 and T5, $n = 29$ participants had no data at T4). Source data are provided as a Source Data file.

adolescents when using the longer time frame of 365 days, and 220 days for the shorter time frame of 220 days. The half-life estimates of our main analysis were comparable to those of the sensitivity analysis which supports robust findings covering different infectious VOCs. The estimates of the shorter time frame were comparable to studies in adults reporting between 145 to 238 days[27–32]. Data on anti-spike IgG half-life in children and adolescents is limited and results are inconclusive, ranging from faster[32] to similar[25] or even slower[23,26] decay of anti-spike IgG antibodies comparing children and adults. The half-life estimates of our longer time frame (365 days) were higher than what has previously been reported in adults[27–32]. Studies with shorter follow-up may have covered primarily the early time after infection in which a faster decay of antibodies takes place, while the longer time frame covered a full year including both the initial fast decline of antibodies followed by a period in which the decay was much slower and steadier[25,32]. Besides the timing, several additional factors may explain differences in anti-spike IgG half-life including the unknown timepoint of infection in our study, missingness of data due to children and adolescents getting vaccinated, as well as varying assumptions employed in the analyses. Furthermore, the persistence and half-life estimates may be influenced by the presence of additional immune responses specific to children and adolescents including mucosal antibodies, T-cells or acute phase proteins[33,34]. Our sample comprised predominantly healthy school children and adolescents, and our data suggest a homogenous antibody decay (Supplementary Fig. 4a–d). Whether the longer persistence and half-life of antibodies in children and adolescents compared to adults (also reported by others[22,23,26]) have any clinical relevance in children and adolescents is unknown. Additionally, we do not know whether children and adolescents with abnormally high or low immune responses, for instance, those with severe COVID-19 or more serious chronic health conditions, behave similarly.

Comparing children and adolescents by exposure status, we found that anti-spike IgG antibody titres and neutralising responses were higher in those with hybrid immunity or vaccination compared to those infected by July 2022. This observation was consistent across age groups. Most studies[35–40] found that individuals with hybrid immunity had the highest anti-spike IgG titres and neutralising responses followed by vaccinated and then infected individuals, but most of these studies focussed on adults. A study by Zaballa et al.[7], found that neutralising activity, especially for Omicron BA.1 and even more so for subsequent subvariants (Omicron BA.2, BA.2.12.1, BA.4/BA.5), was much lower in children and adolescents compared to our study despite the use of the same assay[41]. It is possible that neutralisation in our children and adolescents with vaccination and hybrid immunity was higher due to undetected or repeated infections with Omicron which has been shown to boost neutralisation[42], or different timings since the last infection among studies. Yet, any differences between their and our study may also have resulted from systematic differences in study populations, the technical setup or random chance.

In our study, we may have underestimated the proportion of children and adolescents with hybrid immunity and overestimated those vaccinated only, considering that anti-nucleocapsid IgG antibodies wane quickly and are also less expressed among vaccinated individuals[43–45]. We likely missed infections in early 2022 if anti-nucleocapsid IgG antibodies were undetectable despite infection which could indeed have misclassified children and adolescents, potentially leading to higher median anti-spike IgG titres and neutralisation responses than the true estimates in the vaccinated group and thus overestimation of immune responses in the vaccinated group. This potential misclassification does not seem to be of clinical relevance as both groups (i.e., hybrid and vaccinated) showed comparably high anti-spike IgG and neutralising responses. Numerous studies in adults demonstrate that those with high antibody titres and neutralising activity are best protected against developing a severe course of COVID-19[46–48]. As only one adolescent had a hospital stay of less than 24 h likely related to a SARS-CoV-2 infection, it remains unclear whether the findings in adults of higher protection against a severe course of an acute SARS-CoV-2 infection by vaccination and/or hybrid immunity can also be translated to children and adolescents. A study to test this hypothesis in children and adolescents would require an extremely large population-based study, as a severe course of an acute SARS-CoV-2 infection occurs in less than 1% of children and adolescents, far less than observed in adults[49,50] and the incidence of hospitalisations is below 10 per 100,000 in children and adolescents[51,52].

Previous studies have shown that humoral immune responses may vary significantly among children of different ages[7,22,25,53,54] or compared to adults[7,26,54], but usually show that children, compared to adults, show equal or higher persistent humoral responses and neutralisation activity to SARS-CoV-2 infections are maintained for at least 6 months[22–26]. Based on reports from the literature that indicate potential variations in humoral immune responses among children of different ages, we stratified all analyses by age (<12 or ≥12 years) and found - in agreement with others[53,55]—no difference among age groups, regardless of exposure status. We do not know to what extent small sample sizes, severity of disease, use of different assays or other unknown factors may have contributed to the variability in results. In general, children compared to adults show a higher local interferon response in airways, increased activation of neutrophils and more recruitment of monocytes, dendritic cells, and neutral killer cells at the infection site combined with a lower memory-based cytotoxic immune response to the SARS-CoV-2 virus, and a higher T-cell receptor repertoire diversity able to directly react against SARS-CoV-2 antigens[33,34,56,57]. In addition, there seems to be an immunological cross-protection of seasonal coronaviruses that is more prevalent in children[26,58,59]. All these mechanisms may contribute to the milder clinical affection in children of all ages[34,49].

SARS-CoV-2 reinfections were rare before the appearance of Omicron and increased significantly thereafter in all populations including children[22,60,61]. While infections in children and adolescents are typically mild or asymptomatic, reinfections in children and adolescents seem to be even milder than primary infections[61]. Nevertheless, reinfection rates in our cohort during the Omicron wave were higher in infected (i.e., vaccine-naïve) children and adolescents than in the vaccinated counterpart (especially in those with recent vaccination), but none of the children and adolescents were hospitalised due to a SARS-CoV-2 infection during the Omicron wave. Studies in adults have shown an increased risk of post-COVID-19 condition after SARS-CoV-2 reinfection even in those vaccinated[62]. Whether this increased risk for post-COVID-19 condition with reinfections in the adult population is transferable to children and adolescents is not known. A better understanding of post-COVID-19 conditions in children and adolescents by population-based studies with the monitoring of reinfection rates and trials on vaccine safety among the growing population will inform vaccine policies in children and adolescents in the future.

The Ciao Corona study is one of the few large longitudinal studies in children and adolescents[24,25,63]. We were able to capture the time periods of the circulation of major SARS-CoV-2 variants (Wildtype, Delta, and Omicron BA.1) during our five testing rounds between June 2020 and July 2022. Serological testing allowed us to detect children and adolescents with SARS-CoV-2 infection irrespective of symptomatology. With the longitudinal cohort, we were able to assess temporal changes in humoral immune responses over the first two years of the COVID-19 pandemic, considering infections, vaccinations, and their combinations.

However, some limitations need to be considered when interpreting the findings of this study. First, the exact timing of SARS-CoV-2 infections in children and adolescents is not known in sero-

epidemiological studies. Thus, infection could have occurred days to months before a participant tested seropositive in our study. Second, we may have misclassified some children and adolescents when classifying them according to exposure status due to several reasons. The production of antibodies in response to a SARS-CoV-2 infection is not guaranteed. This reduced likelihood of seroconversion is more common in children than in adults[64]. Vaccination status was self-reported by the study participants or their parents and may have been subject to recall bias. This could have led to an over- or underestimation of seroprevalence and differences in antibody titres among the exposure status groups. We were able to validate self-reported vaccination status in 74% of children and adolescents, by agreement between parental questionnaire information and participants' self-report during testing at school. For the remaining 26%, we assessed vaccination status only from children and adolescents during testing at school. Although we cannot deny misclassification, the accuracy of self-reported vaccination for current and prior season vaccinations is high, even in children[65]. Further, differentiation between children and adolescents with hybrid immunity or vaccination only was based on the presence of SARS-CoV-2 anti-nucleocapsid IgG antibodies. Since the anti-nucleocapsid IgG antibodies wane quickly[45] and the response is weaker when vaccinated (as shown in different studies[43–45]), we likely underestimated the number of children and adolescents with hybrid immunity and overestimated those vaccinated. This may have led to higher median anti-spike IgG titres and neutralisation responses than the true estimates in the vaccinated group. Thus, we cannot fully exclude that our findings of similar anti-spike IgG titres and neutralisation response among the vaccinated and hybrid groups are valid results or a consequence of misclassification within the vaccinated group. Third, the estimation of anti-spike IgG half-life bears the limitations that the decay was calculated using the first seropositive result and the time between our testing rounds varied between 4 to 8 months. Due to missing information on the exact timepoint of infection, we set the peak at the first seropositive result to evaluate the decline in anti-spike IgG. However, this approach may possibly underestimate peak antibody titres and enhance the variability of measured values. Consequently, we may have overestimated the half-life in children and adolescents. Fourth, the persistence of anti-spike IgG antibodies over 24 months in 69% of children and adolescents may have been underestimated due to false-positive serological results at T1 (Jun/Jul 2020) when SARS-CoV-2 prevalence was low[66]. Fifth, we did not consider any data on short- or long-term symptoms related to the SARS-CoV-2 infection in children and adolescents, due to inconsistent reporting and potential confounding with other infections and vaccinations.

In conclusion, we highlighted the importance of serological studies, especially with a longitudinal design, as a COVID-19 monitoring tool and the development of humoral immune responses in children and adolescents. Our findings show that the Omicron wave and the rollout of vaccines led to almost 100% seropositivity and boosted seroprevalence and anti-spike IgG antibody titres (by infection, reinfection, and/or vaccination) as well as neutralising activity in children and adolescents. Hybrid immunity and immunity after vaccination induced the highest antibody concentrations as well as neutralisation and are likely to confer the best protection against reinfections and possibly severe disease. During the entire study period, the parents of one adolescent reported a hospital stay of less than 24 h possibly related to an acute SARS-CoV-2 infection.

## Methods
### Study setting and design
The Ciao Corona study is embedded in a nationally coordinated research network *Corona Immunitas* in Switzerland[67]. The protocol of the study was registered prospectively (ClinicalTrials.gov identifier: NCT04448717)[68], and seroprevalence results of the first four Ciao Corona testing rounds can be found elsewhere[69–72]. This repeated cross-sectional analysis is based on a prospective cohort study, using data from children and adolescents who participated at multiple timepoints. We built our longitudinal cohort of children and adolescents that participated at T5 (Jun/Jul 2022) as well as any three or more previous testing rounds. The study took place in the canton of Zurich with around 1.52 million (18% of the Swiss population) ethnically and linguistically diverse inhabitants and comprises rural as well as urban regions. The Ethics Committee of the Canton Zurich, Switzerland, approved the study (BASEC Registration No. 2020-01336). Children provided oral and parents or caregivers written informed consent prior to study enrolment.

The primary outcomes were the longitudinal development of the anti-spike IgG antibodies and neutralising antibody responses against SARS-CoV-2 in school-aged children and adolescents over time. The secondary outcomes were the persistence of antibodies and variation of antibody levels in individuals only infected, vaccinated or with hybrid immunity during the early Omicron period.

In March 2020, the first restrictions and preventive measures were announced by the Swiss Federal Office of Public Health. Schools closed on 16 March 2020 and partially reopened on 10 May 2020, with a combination of in-person and online teaching. On 7 June 2020, schools resumed usual in-person teaching with certain preventive measures (e.g., contact tracing systems within schools, mandatory face masks for school personnel, distancing regulation). Implementation of restrictions varied across schools. For adolescents of 12 years or older, masks were mandatory starting from October 2020 and for children between 9 to 11 years starting from January 2021. This was implemented due to an increase in the incidence of SARS-CoV-2 infections, signalling a second pandemic wave. Throughout the summer of 2021, masks were no longer mandatory for children and adolescents. However, in the canton of Zurich, they were reinstated for all school children and adolescents from December 2021 to mid-February 2022 during the first peak of the Omicron wave[73]. Adolescents of 16 years or older were allowed to get vaccinated starting from May 2021, adolescents between 12-15 years of age since mid-June 2021 and children between 5 and 11 years of age from January 2022[15].

### Population
We randomly selected primary schools in the canton of Zurich and invited for each primary school the secondary school that was the closest geographically. The number of invited schools per district corresponded to the population size of the 12 districts. Out of 156 invited schools, both public and private (around 10%), 55 schools agreed to participate. Classes were randomly selected and stratified by school level: grades 1–2 (6 to 8 years old children) of lower school level, grades 4–5 (9 to 11 years old children) of middle school level, and grades 7–9 (12 to 14 years old adolescents) of upper school level. All children and adolescents in the randomly selected classes were eligible to participate in any of the testing rounds, irrespective of whether they participated at baseline.

### Timeline of testing
In all five testing rounds, we collected venous blood samples at randomly selected schools located in the canton of Zurich. The first testing round (T1) was performed in June/July 2020, the second (T2) in October/November 2020, the third (T3) in March/April 2021, the fourth (T4) in November/December 2021 and the last fifth (T5) testing round in June/July 2022. As shown in the study participant flowchart (Supplementary Fig. 1), we followed corresponding repeated cross-sectional cohorts. The longitudinal cohort consisted of children and adolescents participating in the last (T5) and at least three previous testing rounds.

## Serological testing and neutralisation assay

In all five testing rounds, we visited children and adolescents in schools and collected venous blood samples in K2-EDTA vacutainer tubes (BD). After blood centrifugation, plasma aliquots were stored at −20 °C prior to IgA and IgG antibody analyses. To detect SARS-CoV-2 specific antibodies against spike and nucleocapsid proteins we used the Sensitive Anti-SARS-CoV-2 Spike Trimer Immunoglobulin Serological (SenAS-TrIS) test[74]. In this assay, MagPlex beads were covalently coupled with either the SARS-CoV-2 Spike protein trimer or Nucleocapsid protein using a Bio-Plex 356 Amine Coupling Kit (Bio-Rad, Catalogue 10000148774) per manufacturer's protocol. To each well of Bio-Plex Pro 96-well Flat Bottom Plates (Bio-Rad) diluted protein-coupled beads were added and washed with PBS on a magnetic plate washer (MAG2x programme). Next, 50 µl of individual plasma samples diluted 1:300 in PBS were added to the wells. A pool of pre-COVID-19 pandemic healthy human sera was used as a negative control (BioWest human serum AB males; VWR). Plates were incubated for 1 h at room temperature while shaking, washed with PBS and incubated with 50 µl of a 1:100 dilution of polyclonal Goat F(ab')2 anti-human IgA-PE (for anti-IgA assay; Southern Biotech; Catalogue 2052-09) or polyclonal Goat anti-human IgG-PE (for anti-IgG assay; OneLambda, Catalogue LS-AB2) secondary antibody at room temperature for an additional 45 min while shaking. After incubation, samples were washed with PBS and resuspended in the reading buffer and read on a Bio-Plex (Luminex) 200 plate reader with Bio-Plex Manager software (version 6.2; Bio-Rad) to obtain a mean fluorescence intensity (MFI) value for each sample. The MFI value for each plasma sample was divided by the mean value of the negative control samples to yield an MFI ratio. Test results were considered seropositive if MFI ratios were equal to or above the cutoff of 6 for both anti-spike IgG and anti-nucleocapsid IgG, explaining the 99.2% test specificity and 94.0% test sensitivity[74]. The MFI ratio cutoff of 6 was determined based on negative control samples and samples from SARS-CoV2 PCR-positive donors[74]. As the performance of the assays, used in our study, was found to be stable with only a minor decay up to 8 months after infection[75], we did not control for sensitivity in the decay of time.

We used a cell-free and virus-free assay to detect SARS-CoV-2 neutralising antibodies against the Wildtype SARS-CoV-2, Delta, and Omicron BA.1 variants[41]. For this assay, cryopreserved plasma samples were thawed and diluted in PBS (1:10, 1:30, 1:90, 1:270, 1:810, and 1:2430). Next, 50 µl of each dilution were incubated while shaking for 60 min at room temperature with Luminex beads covalently coupled to the original SARS-C2oV-2 Spike protein (2019nCoV) and Spike variants Delta (B.1.617.2) and Omicron (B.1.1.529) in Bio-Plex Pro 96-well Flat Bottom Plates (Bio-Rad). Negative control on each plate consisted of beads only and dilutions of pooled, pre-COVID-19 pandemic healthy human sera (BioWest human serum AB males; VWR). As a positive control, we included a high concentration (>1 µg/ml) of two broadly neutralising human monoclonal antibodies binding distinct epitopes on the SARS-CoV-2 S protein (Clones P2G3 and P5C3), isolated from previously infected and vaccinated donors. After incubation, the angiotensin-converting enzyme 2 (ACE2) mouse Fc fusion protein (produced by the École Polytechnique Fédérale de Lausanne (EPFL) Protein Production and Structure Core Facility) was added to each well at a final concentration of 1 mg/µl and agitated for an additional 60 min. Beads were washed with PBS on a magnetic plate washer (MAG2x programme) and 50 µl polyclonal Goat F(ab') anti-mouse IgG-PE secondary antibody (Invitrogen, Catalogue 12-4010-87) was added at a 1:100 dilution. Plates were incubated for 45 min at room temperature while shaking, washed with PBS and resuspended in 80 µl reading buffer and analysed on a Bio-Plex 200 plate reader with Bio-Plex Manager software (version 6.2; Bio-Rad)[41].

To measure the binding capacity of our samples to the receptor binding domain of the trimer spike protein of the different SARS-CoV-2 variants, we used averaged MFI values of beads alone without plasma as the 100% binding signal for the ACE2 receptor to the bead-coupled spike trimer and MFI values from wells containing commercial anti-spike blocking antibodies as the maximum inhibition signal[41]. Percent blocking of the spike protein trimer:ACE2 interaction of our samples was calculated using the formula: $\%inhibition = 1 - ([MFItestdilution - MFImaxinhibition]/[MFImaxbinding - MFImaxinhibition])*100)$. A lower limit half maximal Inhibitory concentration (IC50) serum dilution of 50 was set as the specificity cutoff using IC50 values of 104 pre-pandemic healthy donor samples ($cutoff 50 = 12.5 meanIC50 + 4*9.0 standarddeviation(SD)$) to minimise the detection of false-positive samples. Hence, IC50 values of 50 or higher are defined as positive test results[41].

Neutralising activity values were capped at 2430, because it was the value of the highest dilution (1:10, 1:30, 1:90, 1:270, 1:810, and 1:2430) of the Luminex assay[41] and values above 2430 are not meaningful since they are extrapolated.

## Questionnaire

Online questionnaires were sent to participants at enrolment and repeatedly every 3 to 6 months over the duration of the study, collecting information on sociodemographic characteristics, chronic conditions, and vaccination status. For data collection purposes we used the Research Electronic Data Capture (REDCap) platform (current version 10.6.13)[76]. The vaccination status of children and adolescents was either self-reported by children and adolescents on the day of testing in schools or reported by parents/caregivers in online questionnaires. 69% of reports about vaccination were provided by parents via questionnaires and cross-validated with the self-reported data from children and adolescents, 5% were obtained by interviewing parents or caregivers by phone, to address any discrepancies and the remaining 26% were self-reported by the children and adolescents at the day of testing.

Information on hospitalisations was based on reported symptoms potentially, but not exclusively related to a SARS-CoV-2 infection and which were not related to chronic disease or known allergies. If a hospitalisation was reported, we contacted the parents or caregivers by phone to assess why children and adolescents were hospitalised.

## Groups of children and adolescents according to seropositivity and exposure status

To assess the evolution of anti-spike IgG and neutralising antibody titres, we categorised children and adolescents from the longitudinal cohort into four groups according to their vaccination and infection status. Children and adolescents never testing positive for anti-spike IgG were categorised as *negative*, unvaccinated children and adolescents ever testing positive for anti-spike IgG as *infected*, vaccinated children and adolescents testing negative for anti-spike IgG prior to vaccination and never testing positive for anti-nucleocapsid IgG as *vaccinated*, and children and adolescents testing seropositive before getting vaccinated, or when vaccinated and tested positive for anti-nucleocapsid-IgG antibodies as *hybrid*.

## Statistical analysis

We performed descriptive analysis for participants' characteristics and antibody titres, by reporting median (interquartile range) or count (percentage). Neutralising activity was visualised using a log10-transformation of scales. The Wilson method was used to calculate 95% confidence intervals (95% CI) of proportions[77]. We divided the study population into children being younger than 12 years and adolescents of 12 years and older, based on different vaccination policies for younger and older children and adolescents in Switzerland[15].

For the purpose of interpretation, we provide all the MFI values as WHO units per millilitres (U/ml) by using the Elecsys Anti-SARS-CoV2 immunoassay developed by Roche[78]. The Department of Clinical Immunology & Allergy of the University Hospital of Lausanne used

population-based samples to provide the conversion formula of Roche $anti-spikeIgG = 10^{(-0.6108069 + 2.0072882 \times \log10(MFI+1))}$.

We used Bayesian logistic regression to estimate the seroprevalence with 95% credible intervals (95% CrI), using a model which accounts for the sensitivity and specificity of the SARS-CoV-2 antibody test and the cohort's hierarchical structure. The Bayesian approach also allowed to adjust for the geographic district of the school, sex, and school grade of the child and included random effects for school levels (lower, upper, and middle). We used poststratification weights to adjust for the population size of the particular school level and the geographic district. Further details regarding the Bayesian model and weighting approach can be found in the supplementary method and elsewhere[69,79].

Seroprevalence in the first three rounds (T1 to T3) was only referring to unvaccinated children and adolescents since vaccination was only available since June 2021 (rounds T4–T5, see Supplementary Fig. 2). For T4 and T5, we conducted the analysis of seroprevalence for two subgroups: (a) the unvaccinated children and adolescents, and (b) all participating children and adolescents.

To detect and quantify infections and reinfections among children and adolescents between T4 and T5, we used an a priori but arbitrary threshold of 25% or higher increase in anti-spike IgG titres. We judged this threshold to reflect a relevant increase consistent with an infection (in the absence of vaccination), based on observations from several population-based cohorts over the course of the pandemic, and considering any possible measurement error. We then conducted sensitivity analyses using a 15% and 35% threshold.

To determine anti-spike IgG antibody decay times, we included all participants from the longitudinal cohort that seroconverted at any testing round and of whom at least one follow-up serology was performed. We excluded (a) individuals who never tested seropositive for anti-spike IgG antibody, (b) those who had no follow-up assessment after testing seropositive, (c) those who were vaccinated and (d) those with potential reinfection, defined by the presence of anti-nucleocapsid IgG or any increase in anti-spike IgG titres between two testing points. To estimate the slope of antibody decay, we limited the data to the first seropositive result (the closest and therefore likely highest MFI ratio after infection) and all following timepoints, and then realigned the time axis to begin at the first seropositive result for each individual as done by others[31,80,81]. We then fitted the univariable mixed-effects linear decay model for the natural logarithm of the titres, with random intercepts for each participant. We used the formula $(\ln(0.5)/\beta)$ to estimate the half-life ($\lambda$) in days, with $\beta$ being the coefficient for the time deriving from the fitted model[31,80,81]. In the main analysis, we estimated the anti-spike IgG half-life in children and adolescents considering the longest possible time frame of 365 days and a shorter frame of 220 days, to ensure comparability with other published studies[27–32] and also due to the timing of our testing rounds. In a sensitivity analysis to potentially reduce selection bias, we also estimated the half-life (for the longer and shorter time frame as described above) for children and adolescents participating in any four or more testing timepoints regardless of whether they participated at T5.

The analyses were performed with R programming language (v4.2.1)[82], using the tidyverse (v1.3.2), lmerTest (v3.1-3), epitools (v0.5-10.1), lubridate (v1.8.0), janitor (v2.1.0), openxlsx (v4.2.5), broom.mixed (v0.2.9.4) packages, including the RSTAN package (v2.26.16) to fit the Bayesian models[83] (see Supplementary Software). Results were visualised using the ggplot2 (v3.3.6), scales (v1.2.1), cowplot (1.1.1) and RColorBrewer (v1.1-2) packages.

## Reporting summary
Further information on research design is available in the Nature Portfolio Reporting Summary linked to this article.

## Data availability
All data supporting the findings of this study are available within the paper and its supplementary information files. Source data are provided in this paper.

## Code availability
The analysis code used for this study (R programming language) can be found in the Supplementary Software file.

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

## Acknowledgements

We thank Jan H. Schlegel for his contribution to data and code management. The Ciao Corona study was embedded in the nationally coordinated research network *Corona Immunitas*, coordinated by the Swiss School of Public Health (SSPH+). The Ciao Corona study was funded by fundraising of SSPH+, which included funds of the Swiss Federal Office of Public Health as well as private funders (the SSPH+ ethical guidelines for funding will be considered), by the Cantons of Switzerland (Vaud, Zurich, and Basel), by institutional funds of the Universities and by the EU-grant CoVICIS (HORIZON-HLTH-2021-CORONA-0, Project ID 101046041). Additionally, the University of Zurich Foundation provided funding specific to this study. DM received funding from the University of Zurich Postdoc Grant, grant no FK-22-053. The funders played no part in neither the planning and implementation of the study; nor the collection, management, analysis, and interpretation of the data; nor the writing, reviewing, and approving of the manuscript; nor the choice to submit the manuscript for publication. All authors were able to fully access the output of the data analysis and take responsibility for integrity and accuracy.

## Author contributions

S.K. and M.A.P. conceived the study. S.K., M.A.P., T.R. and J.F. developed the preliminary design. S.K., M.A.P., T.R. and A.U. established the study design and methodology. S.K., A.U., T.R., S.R., S.R.H. and A.R. performed participant recruitment, data collection and management. S.R.H., S.R., A.U., D.M., T.B. and A.R. conducted the data cleaning and the statistical analysis. G.P., C.F., C.P. and D.L.C. devised the serology analysis protocol and supervised, conducted and assessed the serological examinations. A.R. wrote the first draft of the manuscript. All authors were involved in the interpretation of the findings, the review and authorisation of the manuscript for intellectual accuracy. S.K. is the corresponding author and guarantor, assuming complete accountability for the conducted research. Furthermore, S.K. had full access to the data and made the final decision to publish. The corresponding author (S.K.) attests that all listed authors meet authorship criteria and that no others meeting the criteria have been omitted. All contributing authors approved the submitted manuscript.

## Competing interests

The authors declare no competing interests.
