## [Peer Review File · Nature Communications]

Persistent humoral immune response in youth throughout the COVID-19 pandemic: prospective school-based cohort studyREVIEWER COMMENTS

Reviewer #1 (Remarks to the Author):

I read with great interest such an important and well done study.

Overall, the approach of the study looks appropriate, in terms of organization of follow-up, laboratory analyses, data collected, despite the mentioned limitations. The study has important implications, however I think their findings can be discussed more in details as they open new perspectives on both the need of vaccinating and not vaccinating children for covid.

I was not able to find the questionnaire the authors mention in table 1, so I could not evaluate its appropriateness and further limitations.

I dont have significant comments on the results, but would rather comment on the discussion of the results.

1- the authors mention that the swiss government closely monitor their data to provide decisions, which is good, and that meetings are periodic since decicions and evidence are dynamic. Due to their data they mentioned to stop vaccination in under 16, however I am unsure that their findings are strongly associated to this message. I want the authors to discuss more on these points:

- vaccinated/hybrid children have higher cross variants protection and were less re-infected (by a large amount). what is the take home message on that? reinfection can lead to long covid for example, as showed by a recent paper by longcovidkidsUK.

- IgG decay of 1 year. Therefore, would you comment about possible need of yearly boosting?

- while many of this 6-16 cohorts have immunity now, what about the younger ones and mostly the newborns that are eligible since 6 months of age?

- I have found not details about the 4 hospitalized children. the authors should provide a supplementary table with clinical details

- did the authors collected the risk of persistent symptoms in those infected with or without previous vaccine/hybride immunit? if not, this should be discussed as a limitation that may question their take home message about no further need of vaccination

- did the authors collected information about vaccine adverse events in the quesitonair?

Reviewer #2 (Remarks to the Author):

This work presents the analysis of cross-sectional and longitudinal anti-SARS-CoV-2 antibody measurements in a school-based cohort in the canton of Zurich, Switzerland, throughout the COVID-19 pandemic. The study shows seroprevalence evolved during the pandemic, as well as how neutralizing capacity evolved before and after the Omicron variant waves in late 2021 to early 2022. Reported findings include that seroprevalence in schools markedly increased during the pandemic; that antibodies remain detectable in children more than 18 months after infection; and that neutralizing capacity against different variants depends on infection and vaccination history, with hybrid immunity providing the strongest neutralizing capacity.

The value of this study relies in the large sample size in comparison to other school-based cohorts in serological surveys, in particular with respect to longitudinal surveys. In this respect the study has potential in supporting previous findings in the literature concerning all three conclusions stated above. As detailed below, I believe that the main relevance of this work lies in the large school-aged longitudinal cohort combining binding and neutralizing antibody measurements. There are however currently significant methodological limitations which may limit its relevance in its

current form for a journal with Nature Communications' readership. I hope that the comments below may give directions in which to make the most out of the data presented in the study.

Major points

- Focus on the longitudinal cohort: The authors currently frame the study in terms of both the seroprevalence and the longitudinal dynamics of anti-SARS-CoV-2 antibodies in school-aged populations. As stated by the authors, the former question has already largely been answered in published studies through repeated cross-sectional surveys internationally, including elsewhere in Switzerland (Zaballa et al., 2023). I believe that the main value of the study is in the longitudinal cohort, although only a fraction of participants had repeated serologies, as relatively few studies have addressed the question of children binding and neutralizing antibody dynamics in time. I would therefore recommend to frame the study on this aspect of the data, with a more in-depth exploration of the results. I realize this would constitute an important change in the presentation of the results, but a possibly smaller one in the narrative as it currently stands.
- Stratification of results by age: Previous studies have shown that seroprevalence, neutralizing antibodies, and antibody kinetics may vary significantly with age in children (eg. Han et al. 2022, Dowell et al. 2022). I would therefore suggest to stratify and discuss the analyses throughout the main text as done in Supplementary Table 2 (< or > 12 years). This would be particularly useful to interpret figure 4, for which it is not clear to me if participants in each subgroup in each panel are comparable in terms of age compositions.
- Definition of groups and interpretation of results: In the limitation paragraph of the Discussion the authors mention that anti-N based categorization of infection may have biased classification from hybrid immunity to vaccination due to decaying anti-N response and lower anti-N response after vaccination (l. 349). Mentioning the relative importance of these two seems quite important to me. In particular giving numbers on decay times from existing studies on the time-varying sensitivity of the immunoassay used in this study if any, are their eventual absence as additional source of uncertainty. The authors mention reports of similar levels of neutralization between vaccinated and hybrid immunity, although others have found significant differences between the two (Frey et al. 2022, Zaballa et al. 2023). I would also ask the authors to discuss more the impact of this miss-classification on the overall results and main messages of the paper. In particular, miss-classification may lead to over-estimation of the antibody titers and neutralization capacity in the "vaccinated" class, thus potentially undermining one of the important points of the study (hybrid immunity and vaccination had similar antibody/neutralization levels).
- Comparability of groups in Figure 4 and 5: Related to the previous point, it is unclear to me how comparable groups are in terms of the factors that may affect binding and neutralizing antibody levels. For instance, it is unclear to me how the Vaccinated T5 can have what seems to be higher neutralizing antibody levels than Vaccinated T4 in the absence of intervening boosting events. I would suppose this is due to time since vaccination, but am unsure if other factors could explain things. As the authors compare groups in terms of distributions and statistics, I would encourage the reporting of all factors that may impact their comparability, including times since vaccination, age and sex. It was not clear whether information on self-reported tests and/or symptoms was implemented in questionnaires, but if so these would also be important to account for. Alternatively, I could suggest performing a regression analysis to account for the contribution of different factors in a systematic way (eg. Zaballa et al. 2023).
- Vaccination status: It is stated that the "Vaccination status of children and adolescents was either self-reported by children and adolescent on the day of testing in schools or reported by parents/caregivers in online questionnaires. ». The reliability of self-reported vaccination status in children and adolescents is questionable. Please provide information on the fraction of data coming from self-reporting and parent/caregiver questionnaires, and if possible an assessment of the reliability of self-reporting in the Discussion.
- Estimation of antibody titers by origin: From what I understand all samples were also tested for anti-N antibodies. Given the type of mRNA-based vaccines available in Switzerland this would enable inferring antibody origin (infection vs. vaccination) by jointly accounting for both antibody types (eg. Zaballa et al. 2023). If feasible I would encourage the authors to add this analysis which would add significantly to the seroprevalence result sections.
- Estimation of binding antibody decay times: The computation of the antibody half-life is based on a subsample of participants of the longitudinal cohort which consists of infected-only participants with no history posterior history of infection as quantified by anti-N serology. Could the authors

comment whether this could introduce a form of selection bias for participants with particular immune responses which could affect the representativeness of the estimate? Regarding the sensitivity analysis, were the two different subsets of participants comparable in terms of age/sex? Another point regards the selection of the longitudinal cohort for the decay time analysis. Were participants with three time points, but no data at T5 removed from the analysis, as the description in the methods seems to imply? If so, please give a rational for this, and clarify the specific set of participant retained by providing data on the number of participants with different number of repeated samples.

- Model equations and code sharing: For clarity, please add all model equations used in the study in an appendix. I would also urge authors to add a public repository with analysis code used in this study as well as minimal datasets to illustrate their use.
- Participation rates: Please provide individual-level participation rates in the study.

Other points of importance:

- Title: The word "immunity" suggests that the study address protection, which this analysis does not answer in its current form. I would therefore suggest to replace with something that indicates detection through measurements.
- Omicron sub-variants: Given the emergence of Omicron subvariants and their distinct characteristics, systematically mentioning what subvariants were analyzed in this study would better enable to contextualize the study in the existing literature.
- Figure 3: It is very hard for me to make sense of the longitudinal trajectories with these visualization, which is the main aim of the figure from what I understand. I would suggest following common practice in representing longitudinal antibody trajectories using dots and lines per participant, with the x axis being the time of sample collection (eg. Figure 1 in Iyer et al. 2020).
- Figure 4/5: It is unclear to me whether T4 Infected/T5 Infected are participants with infections before T4 AND occurring between T4 and T5, or whether it is not possible to differentiate this here. Please add this information to the caption/discussion.
- Results in Supplementary table 6: It is unclear for me how to read the table. In particular, how can the column anti-N negative have numbers for percentage increase/decrease? And how was the 25% threshold determined? I believe that this is an important part of the results, but would require more explanation to value it in the main text. (Also note the typo in the caption referring to "older infection").
- Comparison of neutralization capacity results with Zaballa et al. 2023: In Zaballa et al. 2023, much lower neutralization capacity among school-aged children were found using the same neutralizing test, in particular below 12 years of age, than in this study. Could the authors comment on the potential reasons these differences may exist, and their implications for the interpretation of this study's results? I encourage the authors to provide more details (equations, code, etc) for future comparison of results between these two studies.
- Tone of results/discussion/conclusion: I invite the authors to tone down the novelty and importance of results throughout the text as these are not warranted in my opinion, and to leave more space for the reader's interpretation and evaluation.

In order of appearance:

- L. 66-68: the message conveyed by this sentence is not clear to me
- L. 71: 5 hospitalizations (even short) – or 3 as stated line 119? - in a sample of healthy children and adolescents seems high, and the word "only" seems inappropriate to me in this statement. Please provide contextual data on hospitalization rates in children/adolescents for SARS-CoV-2 infections in Switzerland or internationally.
- L. 86-87 (ref 1-6): Multiple seroprevalence studies were done in Switzerland, including the Omicron period which provide relevant context to this study (eg. Zaballa et al. 2023). If the authors chose to exclude these studies from the reference list please provide a rational for doing so.
- L. 103-105: Please add a reference to this statement.
- L. 107: the phrase "longitudinal development" is vague given that it could refer to the

longitudinal cohort only. I would recommend keeping the word longitudinal for serial participant-level measurements and using "repeated cross-sectional" for the time evolution of seroprevalence. L. 114: "between 1876 and 2500" please check consistency with Figure 1/Table 1 which say 1875.

- Figure 2. Line "unvaccinated" in red and not light blue as indicated in legend.
- L. 163-169: In these results of antibody detection in time, how were possible re-infections accounted for? Credible intervals should be provided, in particular for the last two groups which had very limited sample sizes.
- L. 171-173: Please provide quantitative support for these statements.
- L. 174: It seems misleading to speak of "duration of protection" here, although anti-S IgG have indeed been shown to be correlates of protection in some cases. Please nuance this statement.
- L. 202: Cf. Major points. How can vaccinated-only participants increase in both binding and neutralizing antibody levels between T4 and T5? Please develop in the discussion.
- L. 211-214: This seems to be an important point. Cf. Comment in other important points on Table S6. To dig this further, could a sub-analysis also be done using binding/neutralization capacity as exposure variable?
- L. 272-272: Given the number of studies on school-aged seroprevalence since the Omicron subvariants I think that "remarkable" is somewhat of an overstatement.
- L. 276-278: "These findings are consistent with other international studies in children and adolescents also reporting high seroprevalence and titre levels by mid 2022". It seems relevant to include a comparison to results from Switzerland here (Zaballa et al. 2023). As noted above, please provide a comparison of results and discussion of differences.
- L. 292: See Dowel et al. for a comprehensive analysis of anti-SARS-CoV-2 immune response in children.
- L. 320: Rates of long-COVID have been reported to be an order of magnitude larger (eg. Dumont et al. 2022), which may be worth mentioning as a public health relevant outcome.
- L. 322-329: All preventive measures were lifted in February/March 2022, and seroprevalence estimates almost reached 100% in July 2022. It is questionable whether this can be considered as a public health success, in particular regarding the risks of long COVID, and the potential longer-term risks of infection. I invite the authors to tone down or remove this paragraph as I do not believe it is suited for a scientific publication in an academic journal.
- L. 335: Was there any information in this study to quantify asymptomatic rates?
- L. 337: I would invite the authors to avoid "first study"-type sentences of this type which do not contribute to the scientific value of the paper.
- L. 392: Please state from which age masks were mandatory. I believe that young children never had to wear a mask at school.
- L. 414: Cf. Point about the definition of the longitudinal cohort with respect to the computation of antibody decay time
- L. 427: Are any studies available on decay of sensitivity with time?
- L. 439: Did questionnaires ask for COVID-19-related hospitalizations and symptoms? If not, how was the information about hospitalization reported in the abstract retrieved?
- L. 467: Was household information available? If so, what was the rationale for not accounting for household clustering? Also, please report posteriors for all inferred model parameters of importance in the supplement. Additionally, please indicate how convergence was assessed.
- Figure 5: In panel (a) it seems like all values for T4 are capped at an upper limit. Does the neutralizing test have an upper limit of quantification? If so, please mention in the methods.
- Supplementary Figure 3/Table 5: Interpretation of Roche-S results is not straightforward to the scientific community at large. I would suggest to transform these results to International Units instead.

References

Doucette, E.J., Gray, J., Fonseca, K., Charlton, C., Kanji, J.N., Tipples, G., Kuhn, S., Dunn, J., Sayers, P., Symonds, N. and Wu, G., 2023. A longitudinal seroepidemiology study to evaluate antibody response to SARS-CoV-2 virus infection and vaccination in children in Calgary, Canada from July 2020 to April 2022: Alberta COVID-19 Childhood Cohort (AB3C) Study. *Plos one*, 18(4), p.e0284046.

Dumont, R., Richard, V., Lorthe, E., Loizeau, A., Pennacchio, F., Zaballa, M.E., Baysson, H., Nehme, M., Perrin, A., L'Huillier, A.G. and Kaiser, L., 2022. A population-based serological study of post-COVID syndrome prevalence and risk factors in children and adolescents. *Nature communications*, 13(1),

Dowell, A.C., Butler, M.S., Jinks, E., Tut, G., Lancaster, T., Sylla, P., Begum, J., Bruton, R., Pearce, H., Verma, K. and Logan, N., 2022. Children develop robust and sustained cross-reactive spike-specific immune responses to SARS-CoV-2 infection. *Nature immunology*, 23(1), pp.40-49.

Frei, A., Kaufmann, M., Amati, R., Butty Dettwiler, A., von Wyl, V., Annoni, A.M., Pellaton, C., Pantaleo, G., Fehr, J.S., D'Acromont, V. and Bochud, M., 2022. Development of hybrid immunity during a period of high incidence of infections with Omicron subvariants: A prospective population based multi-region cohort study. *medRxiv*, pp.2022-10.

Han, M.S., Um, J., Lee, E.J., Kim, K.M., Chang, S.H., Lee, H., Kim, Y.K., Choi, Y.Y., Cho, E.Y., Kim, D.H. and Choi, J.H., 2022. Antibody responses to SARS-CoV-2 in children with COVID-19. *Journal of the Pediatric Infectious Diseases Society*, 11(6), pp.267-273.

Iyer, A.S., Jones, F.K., Nodoushani, A., Kelly, M., Becker, M., Slater, D., Mills, R., Teng, E., Kamruzzaman, M., Garcia-Beltran, W.F. and Astudillo, M., 2020. Persistence and decay of human antibody responses to the receptor binding domain of SARS-CoV-2 spike protein in COVID-19 patients. *Science immunology*, 5(52), p.eabe0367.

Zaballa, M.E., Perez-Saez, J., de Mestral, C., Pullen, N., Lamour, J., Turelli, P., Raclot, C., Baysson, H., Pennacchio, F., Villers, J. and Duc, J., 2023. Seroprevalence of anti-SARS-CoV-2 antibodies and cross-variant neutralization capacity after the Omicron BA. 2 wave in Geneva, Switzerland: a population-based study. *The Lancet Regional Health–Europe*, 24.

REVIEWER COMMENTS

Reviewer #1 (Remarks to the Author):

I read with great interest such an important and well done study.

Overall, the approach of the study looks appropriate, in terms of organization of follow-up, laboratory analyses, data collected, despite the mentioned limitations. The study has important implications, however I think their findings can be discussed more in details as they open new perspectives on both the need of vaccinating and not vaccinating children for covid.

I was not able to find the questionnaire the authors mention in table 1, so I could not evaluate its appropriateness and further limitations.

Thank you very much for your valuable comments.

Please find the part of the questionnaire where we ask for chronic conditions here. The remaining questions in Table 1 are purely descriptive.

Question: Has a doctor or a specialist ever diagnosed your son or daughter with any of the following conditions?

- 1) Yes, asthma.*
- 2) Yes, hay fever.*
- 3) Yes, celiac disease.*
- 4) Yes, lactose intolerance.*
- 5) Yes, allergies (other than hay fever).*
- 6) Yes, eczema/atopic dermatitis in children.*
- 7) Yes, diabetes mellitus (diabetes).*
- 8) Yes, chronic inflammatory bowel disease (ulcerative colitis or Crohn's disease).*
- 9) Yes, high blood pressure (hypertension).*
- 10) Yes, attention deficit (ADHD, ADD).*
- 11) Yes, epilepsy.*
- 12) Yes, joint disease (e.g., arthritis).*
- 13) Yes, depression/anxiety disorder.*
- 14) Yes, other chronic condition(s), namely:*
- 15) I don't know.*
- 16) No.*

I don't have significant comments on the results, but would rather comment on the discussion of the results.

1) the authors mention that the Swiss government closely monitor their data to provide decisions, which is good, and that meetings are periodic since decisions and evidence are dynamic. Due to their data they mentioned to stop vaccination in under 16, however I am unsure that their findings are strongly associated to this message.

We agree with your comment. We were in close contact with the Federal Office of Public Health (FOPH) Switzerland, kept them up to date and informed them regarding our results. But since we do not know what exact decisions were taken based on our data or other considerations, we dropped the respective sentences.

I want the authors to discuss more on these points:

2) vaccinated/hybrid children have higher cross variants protection and were less re-infected (by a large amount). what is the take home message on that? Reinfection can lead to long covid for example, as showed by a recent paper by longcovidkidsUK .

Vaccinated children and adolescents or those with hybrid immunity have higher anti-spike IgG titres and a better neutralising response and become less frequently reinfected compared to only infected children (lines 398-411). However, assessing the association of immune responses with Long COVID was beyond the scope of this paper. Ciao Corona, although being quite a large, prospective cohort study is underpowered to address this question. The take home message of our paper with respect to the concentrations of (neutralizing) antibodies is that similar to adults, hybrid immunity and immunity after vaccination induce the highest antibody concentrations and are likely to confer the best protection against reinfections and possibly severe disease (lines 456-458).

3) IgG decay of 1 year. Therefore, would you comment about possible need of yearly boosting?

This is a very interesting and important question. However, we conducted a population-based study looking at the seroprevalence and development of antibodies in the overall population of children and adolescents and not at effects of boosting. Our data can (and did) help in a specific context and for a specific area (such as was the case for Switzerland) to support vaccination strategies. Yet, we were and are not in the position to make recommendations since this needs to follow a systematic process (e.g., following GRADE) and consideration of a more comprehensive body of evidence including RCTs, observational studies like ours but and also those that look at side effects of vaccination and into the association of immunological and outcome data. Special committees, like the Federal Commission for Vaccination Matters (EKIF) in Switzerland, are in charge of making such recommendations following a structured and multifaceted process.

4) while many of this 6-16 cohorts have immunity now, what about the younger ones and mostly the new-borns that are eligible since 6 months of age?

We do not have any data for this age group (0-5 years) since our study was embedded in the school system that for our study started at grade 1 (≥ 6 years). Therefore, we are not able to make any statement for a younger population.

5) I have found not details about the 4 hospitalized children. the authors should provide a supplementary table with clinical details.

We have implemented your suggestion in the revised version of the manuscript (lines 124-126, 582-585). Please see Supplementary Table 3.

6) did the authors collect the risk of persistent symptoms in those infected with or without previous vaccine/hybrid immunity? if not, this should be discussed as a limitation that may question their take home message about no further need of vaccination

Thank you for raising this point. We did not assess the risk of persistent symptoms in vaccinated or hybrid individuals, since it was beyond the scope of this paper (see comment number 2). We added this to our limitation section (lines 447-449).

In our understanding, we do not provide a take home message about no further need of vaccination, as our study design does not allow to give any recommendation about any further need of vaccination.

7) did the authors collected information about vaccine adverse events in the questionnaire?

We did not collect adverse events after vaccination in the questionnaire as we do not have an appropriate study design to address adverse events after vaccination. Such cohort studies¹ need to enrol participants at the time of vaccination so that the temporal sequence of events is accurate. In our study, children and adolescents have received vaccinations at varying times and with unknown doses, and we do not have data on adverse events during the time period where the vast majority of adverse effects occur following vaccination (i.e., within the first week).

Reviewer #2 (Remarks to the Author):

This work presents the analysis of cross-sectional and longitudinal anti-SARS-CoV-2 antibody measurements in a school-based cohort in the canton of Zurich, Switzerland, throughout the COVID-19 pandemic. The study shows seroprevalence evolved during the pandemic, as well as how neutralizing capacity evolved before and after the Omicron variant waves in late 2021 to early 2022. Reported findings include that **1. seroprevalence in schools markedly increased during the pandemic; that 2. antibodies remain detectable in children more than 18 months after infection; and that 3. neutralizing capacity against different variants depends on infection and vaccination history, with hybrid immunity providing the strongest neutralizing capacity.**

The value of this study relies in the large sample size in comparison to other school-based cohorts in serological surveys, in particular with respect to longitudinal surveys. In this respect the study has potential in supporting previous findings in the literature **concerning all three conclusions stated above**. As detailed below, I believe that the main relevance of this work lies in the large school-aged longitudinal cohort combining binding and neutralizing antibody measurements. There are however currently significant methodological limitations which may limit its relevance in its current form for a journal with Nature Communications' readership. I hope that the comments below may give directions in which to make the most out of the data presented in the study.

Major points

1) **Focus on the longitudinal cohort:** The authors currently frame the study in terms of both the seroprevalence and the longitudinal dynamics of anti-SARS-CoV-2 antibodies in school-aged populations. As stated by the authors, the former question as already largely been answered in published studies through repeated cross-sectional surveys internationally, including elsewhere in Switzerland (Zaballa et al., 2023). I believe that the main value of the study is in the longitudinal cohort, although only a fraction of participants had repeated serologies, as relatively few studies have addressed the question of children binding and neutralizing antibody dynamics in time. I would therefore recommend to frame the study on this aspect of the data, with a more in-depth exploration of the results. I realize this would constitute an important change in the presentation of the results, but a possibly smaller one in the narrative as it currently stands.

Thank you for your constructive comments and for taking the time to review our paper so comprehensively.

Regarding your comment above, we agree that the main value of the study lies in the longitudinal cohort. We have therefore adapted the main body of the manuscript. We now focus predominantly on the longitudinal cohort throughout the manuscript and have moved nearly all of the repeated cross-sectional cohort results to the supplementary material. We only kept some descriptives in the main body of the manuscript that allows the readers to understand how the longitudinal cohort was developed.

2) **Stratification of results by age:** Previous studies have shown that seroprevalence, neutralizing antibodies, and antibody kinetics may vary significantly with age in children (eg. Han et al. 2022, Dowell et al. 2022). I would therefore suggest to stratify and discuss the analyses throughout the main text as done in Supplementary Table 2 (< or > 12 years). This would be particularly useful to interpret figure 4, for which it is not clear to me if participants in each subgroup in each panel are comparable in terms of age compositions.

We agree with your suggestion of stratifying all results by age (<12 and ≥12 years) and have implemented this grouping in the revised version of the manuscript (Supplementary Figure 6a-d and 7a-b; and Supplementary Table 7a-d and 10a-b). Overall, we found no differences in anti-spike IgG titres and neutralising response in younger (<12 years) and older (≥12 years) participants. We have added the two missing references (Han 2022, Dowell 2022) and more studies that looked at differences in antibody responses within children of different age or in comparison to adults. We have also extended the discussion by focussing on age differences in antibody responses and neutralisation (see lines 228-230, 285-287, 380-396, 601-603).

3) **Definition of groups and interpretation of results:** In the limitation paragraph of the Discussion the authors mention that anti-N based categorization of infection may have biased classification from hybrid immunity to vaccination **due to decaying anti-N response and lower anti-N response after vaccination** (l. 349). **Mentioning the relative importance of these two seems quite important to me.** In particular giving numbers on decay times from existing studies on the time-varying sensitivity of the immunoassay used in this study if any, are their eventual absence as additional source of uncertainty.

We agree that it would be nice to have information about the relative importance of the decaying anti-nucleocapsid response versus the lower anti-nucleocapsid response with vaccination. As there is a lack of studies in children and adolescents in this field, we are unable to reason about the relative importance of both factors potentially misclassifying our children and adolescents among the vaccinated and hybrid group. We believe that based on our results of equally high anti-spike IgG with high neutralisation capacity in both groups and the additional analyses based on an increase in anti-spike IgG of 25% and more (see Supplementary Table 8a-b) we have taken the best approach to interpret results. Establishing a hypothetical construct based on our data and extrapolated data in adults would not help to better understand the humoral immunity of children and adolescents and would not change our results of comparable antibody titers and neutralisation in vaccinated children and adolescents and those with hybrid immunity.

Regarding existing studies, our lab team analysed the serology data for 66 SARS-CoV-2 infected donors for whom they had longitudinal data and a reliable date of infection. This study showed that using their Luminex N-protein serology assay, donors remained anti-nucleocapsid IgG antibody positive (>6 ratio) or indeterminate (4-6 ratio) for an average of 6.4 months. In the context of defining a hybrid (vaccinated then infected) versus pure vaccine-induced immune response, 83.3% of infected donors remained positive or indeterminate at 4 months post infection.

A point to consider is that in the scenario above, the infections were in previously vaccinated donors. Since the infection would be less severe and cleared more rapidly in vaccinated compared to non-vaccinated donors, the magnitude and durability of the anti-nucleocapsid IgG specific antibody response and decay time would be reduced significantly in the donors they analysed. In contrast, vaccine naïve individuals that were infected would have a stronger anti-nucleocapsid IgG antibody response in the first months following infection and a longer decay time than 6.4 months until the effects of infection were negative in the Luminex anti-N assay. Indeed, the answer will be different for vaccinated then infected and infected then vaccinated hybrid immunity patients, where the former is 6.4 months and the latter is likely to be several months longer (~9 months). A second point to consider is that they have longitudinal analysis of the vaccinated then infected hybrid immunity donors so the 4 months window for detecting 83.3% of infections by their anti-N assay is quite good. A study looking at performance of the assays used in our study, found those assays to be stable with only a minor decay up to 8 months after infection ².

Nevertheless, these results are based on an adult population with precise knowledge of the time of infection and vaccination. Our study is completely different as it is based on children and adolescents without precise timing of infection nor vaccination and their respective temporal sequence.

Based on all these arguments, we would like to refer to the above mentioned study in the method section and abstain from discussing this point.

The authors mention reports of similar levels of neutralization between vaccinated and hybrid immunity, although others have found significant differences between the two (Frey et al. 2022, Zaballa et al. 2023). I would also ask the authors **to discuss more the impact of this miss-classification on the overall results and main messages of the paper**. In particular, miss-classification may lead to over-estimation of the antibody titers and neutralization capacity in the “vaccinated” class, thus potentially undermining one of the important points of the study (hybrid immunity and vaccination had similar antibody/neutralization levels).

Thank you for this important input and comment. We agree and have extended our part in the discussion regarding the misclassification of vaccinated children and adolescents and its implication on the overall results (lines 363-379, 423-438).

4) Comparability of groups in Figure 4 and 5: Related to the previous point, it is unclear to me how comparable groups are in terms of the factors that may affect binding and neutralizing antibody levels. For instance, it is unclear to me how the Vaccinated T5 can have what seems to be higher neutralizing antibody levels than Vaccinated T4 in the absence of intervening boosting events. I would suppose this is due to time since vaccination but am unsure if other factors could explain things. As the authors compare groups in terms of distributions and statistics, I would encourage the reporting of all factors that may impact their comparability, including times since vaccination, age and sex. It was not clear whether information on self-reported tests and/or symptoms was implemented in questionnaires, but if so these would also be important to account for. Alternatively, I could suggest performing a

regression analysis to account for the contribution of different factors in a systematic way (eg. Zaballa et al. 2023).

This is a very important point. We agree that it is somewhat difficult to interpret the increase between T4 and T5 without further evidence on how other factors such as the exact date of vaccination, time of infection, the temporal sequence of infection and vaccination, sex and age impacted both binding and neutralizing antibody levels. Due to the nature of the study and inconsistent reporting throughout the questionnaires, we do not have the exact date of vaccination nor timepoint of infection, but we were able to divide participants into vaccinated before and/or after T4 (e.g., vaccinated only before OR only after T4, or vaccinated before AND after T4).

We stratified the children and adolescents according to their exposure status (i.e., hybrid, vaccinated, infected, negative) by time of vaccination (i.e., before or after T4 (labelled in the new plot as “older vacc” and “recent vacc”, respectively) as well as acute infection (i.e., newly detectable anti-nucleocapsid IgG before or after T4 (labelled in the plot as “+IgN” when anti-nucleocapsid IgG was detectable). In Figures 1 and 2 and Tables 1 and 2 below, you can see anti-spike IgG titres and neutralising response results in more details across the different groups. As shown in the figures, we were not able to detect a difference in anti-spike IgG titres nor in neutralising response in children and adolescents with a recent vaccination compared to those with an older vaccination. This may be due to the misclassification of children and adolescents from the hybrid into the vaccinated group, since anti-nucleocapsid IgG antibodies wane quickly and the response is weaker when vaccinated. When comparing children and adolescents with hybrid immunity with or without a recent infection and/or vaccination, we found no differences in both anti-spike IgG titres and neutralising response. However, we do see a difference in both anti-spike IgG titres and neutralising activity when comparing infected children and adolescents with and without anti-nucleocapsid IgG (i.e., with or without a recent infection (+IgN)), which can be explained by a recent infection boosting both anti-spike IgG and neutralising antibodies. Nevertheless, the group of infected children and adolescents without anti-nucleocapsid IgG at T5 was very small, and in itself showed a slight increase in anti-spike IgG suggesting that also in this small group, misclassification might have occurred. Based on these findings, we are not confident on reporting any differences among the infected children and adolescents.

Due to the sparse data and lack of information on relevant factors (such as exact time of infection and date of vaccination), it does not seem sensible to us to do a regression analysis in our study, and we therefore prefer to keep this analysis descriptive.

In summary, we agree that it is very important to be as specific as possible. We added the results regarding the (non-existing) difference in titers among children and adolescents with recent and older vaccination (lines 225-228, 282-285), but did not discuss titers and neutralisation in children based on recency of infections in the infected group. We would prefer not to add these figures and tables into the main body of the manuscript nor in the supplement of our paper, since they do not provide additional or new insights/findings beyond on what we report already. In addition, the number of children were also extremely small in many subgroups.

Nevertheless, if the editor considers these figures and tables as important, we will gladly add them to the supplement.

Figure 1: Evolution of anti-spike IgG MFI ratios between T4 (Nov/Dec 2021) and T5 (June/July 2022) among negative, vaccinated children and adolescents, and those with hybrid immunity.

Table 1: Supplementary Table 8: Anti-spike IgG MFI ratios between T4 (Nov/Dec 2021) and T5 (Jun/Jul 2022) among negative, infected, vaccinated children and adolescents and those with hybrid immunity.

n	T4 Nov/Dec 2021	T5 Jun/Jul 2022	T4: Anti-spike IgG Titers (MFI ratio) Median and IQR	T5: Anti-spike IgG Titers (MFI ratio) Median and IQR
12	T4 Hybrid	T5 Hybrid + IgN + older vacc	109.8 (103.9-111.8)	135.4 (122.7-156)
4	T4 Hybrid	T5 Hybrid + IgN + recent vacc	113.7 (106.5-122.7)	126 (123.8-128.3)
6	T4 Hybrid	T5 Hybrid + older vacc	100.2 (89.4-120.9)	141.2 (134-154.8)
7	T4 Hybrid	T5 Hybrid + recent vacc	91.1 (77-108.5)	123.7 (118.2-141.8)
6	T4 Hybrid + IgN	T5 Hybrid + IgN + older vacc	104.5 (90.5-124.8)	122.9 (106.7-140.3)
1	T4 Hybrid + IgN	T5 Hybrid + IgN + recent vacc	116.4 (116.4-116.4)	141.3 (141.3-141.3)
2	T4 Hybrid + IgN	T5 Hybrid + older vacc	77.1 (71.6-82.6)	138.9 (137.1-140.8)
4	T4 Infected	T5 Hybrid + IgN + recent vacc	32.5 (24.1-47.7)	146.4 (133.7-159.6)
21	T4 Infected	T5 Hybrid + recent vacc	31.4 (21.4-40.7)	150.5 (133.5-163.7)
10	T4 Infected	T5 Infected	37.5 (27.5-41.8)	53.1 (42.4-68.6)
79	T4 Infected	T5 Infected + IgN	33.7 (22.2-44.3)	97.7 (73-126.4)
3	T4 Infected + IgN	T5 Hybrid + IgN + recent vacc	44.3 (40-46.5)	152 (134.2-152.6)

5	T4 Infected + IgN	T5 Hybrid + recent vacc	65.7 (64.7-69.4)	142.2 (126-158.4)
7	T4 Infected + IgN	T5 Infected	41.1 (35.9-51.7)	54.2 (30.3-75.1)
47	T4 Infected + IgN	T5 Infected + IgN	47.3 (29-63.8)	89.6 (63.5-116.8)
33	T4 recent Vacc	T5 Hybrid + IgN + older vacc	103.3 (81.1-119)	131.2 (120.5-154.2)
27	T4 recent Vacc	T5 Hybrid + IgN + recent vacc	102.6 (81.6-119.9)	150.1 (127.1-157.7)
47	T4 recent Vacc	T5 recent Vacc	102.8 (80.6-115.4)	128.5 (119-156.1)
37	T4 recent Vacc	T5 older Vacc	113.3 (102.2-129)	125.1 (117.4-151.6)
47	T4 Negative	T5 Hybrid + IgN + recent vacc	1 (1-1)	133.6 (117.2-144.9)
2	T4 Negative	T5 Hybrid + recent vacc	5.3 (5.3-5.4)	161.5 (157.7-165.3)
62	T4 Negative	T5 Infected	1 (1-1.1)	14.9 (10.1-19.2)
148	T4 Negative	T5 Infected + IgN	1 (1-1.2)	43 (22.8-72.1)
73	T4 Negative	T5 recent Vacc	1 (1-1)	126.9 (100-141.4)
1	T4 Negative	T5 older Vacc	1 (1-1)	89.5 (89.5-89.5)

Figure 2: Evolution of neutralizing response between T4 (Nov/Dec 2021) and T5 (June/July 2022) among negative, infected, vaccinated children and adolescents, and those with hybrid immunity.

Table 2: Supplementary Table 8: Neutralising response between T4 (Nov/Dec 2021) and T5 (Jun/Jul 2022) among negative, infected, vaccinated children and adolescents and those with hybrid immunity.

Variant of concern	n	T4 Nov/Dec 2021	T5 Jun/Jul 2022	T4: IC50 Median and IQR	T5: IC50 Median and IQR	T4: % of kids above thres hold	T5: % of kids above thres hold
Wildtype	12	T4 Hybrid	T5 Hybrid + IgN + older vacc	715.3 (465.6-1584.5)	525.5 (358.6-691.5)	100	100
Wildtype	4	T4 Hybrid	T5 Hybrid + IgN + recent vacc	738.8 (661.6-825.9)	1996.3 (1520.6-6669.2)	100	100
Wildtype	6	T4 Hybrid	T5 Hybrid + older vacc	2440 (2427.3-2440)	980.7 (940.4-1162)	100	100
Wildtype	7	T4 Hybrid	T5 Hybrid + recent vacc	500 (303.1-1008.5)	690.6 (284.1-811.4)	100	100
Wildtype	6	T4 Hybrid + IgN	T5 Hybrid + IgN + older vacc	616.3 (250.3-1071.3)	324.7 (176-563.6)	100	100
Wildtype	1	T4 Hybrid + IgN	T5 Hybrid + IgN + recent vacc	2440 (2440-2440)	1080.6 (1080.6-1080.6)	100	100
Wildtype	2	T4 Hybrid + IgN	T5 Hybrid + older vacc	2440 (2440-2440)	458.3 (388.2-528.4)	100	100
Wildtype	4	T4 Infected	T5 Hybrid + IgN + recent vacc	39.7 (31.2-65.6)	602.6 (435.2-693.5)	25	100
Wildtype	21	T4 Infected	T5 Hybrid + recent vacc	40.2 (30.6-63.3)	479.6 (424.4-628.9)	42.9	100
Wildtype	10	T4 Infected	T5 Infected	60.2 (50.1-72)	61.1 (55.1-93.1)	70	90
Wildtype	79	T4 Infected	T5 Infected + IgN	46.8 (29-70.6)	139.7 (95.1-210.3)	48.1	96.2
Wildtype	3	T4 Infected + IgN	T5 Hybrid + IgN + recent vacc	59.2 (52.4-61.6)	290.8 (288.7-395.3)	66.7	100
Wildtype	5	T4 Infected + IgN	T5 Hybrid + recent vacc	84.3 (82.2-86.2)	580 (514.7-687.8)	100	100
Wildtype	7	T4 Infected + IgN	T5 Infected	56.4 (40.6-86.7)	60.9 (39.9-104.4)	57.1	57.1
Wildtype	47	T4 Infected + IgN	T5 Infected + IgN	69.7 (49.1-131.3)	133.7 (70.9-213.5)	72.3	87.2
Wildtype	33	T4 recent Vacc	T5 Hybrid + IgN + older vacc	634.9 (438-809.7)	742.4 (633.7-1552.6)	100	100
Wildtype	27	T4 recent Vacc	T5 Hybrid + IgN + recent vacc	423 (246-619.6)	1490.5 (831.1-2753.8)	100	100
Wildtype	47	T4 recent Vacc	T5 recent Vacc	379.4 (210.5-533.8)	647.7 (384.4-1174.7)	97.9	100
Wildtype	37	T4 recent Vacc	T5 older Vacc	444.6 (313-686.3)	477.3 (146.7-844.6)	100	100
Wildtype	47	T4 Negative	T5 Hybrid + IgN + recent vacc	0 (0-0)	354 (204.8-494.6)	0	93.6
Wildtype	2	T4 Negative	T5 Hybrid + recent vacc	0 (0-0)	303.6 (268.2-339.1)	0	100
Wildtype	62	T4 Negative	T5 Infected	0 (0-0)	10.3 (7.3-16.2)	0	1.6
Wildtype	148	T4 Negative	T5 Infected + IgN	0 (0-0)	35.9 (16.7-76.1)	0	41.2
Wildtype	73	T4 Negative	T5 recent Vacc	0 (0-0)	212.3 (129.6-281.8)	0	91.8
Wildtype	1	T4 Negative	T5 older Vacc	0 (0-0)	77.9 (77.9-77.9)	0	100
Delta	12	T4 Hybrid	T5 Hybrid + IgN + older vacc	332.7 (188.8-488)	245.4 (173.8-301)	91.7	100
Delta	4	T4 Hybrid	T5 Hybrid + IgN + recent vacc	269.9 (249.4-337.3)	997.5 (763.5-1965.7)	100	100
Delta	6	T4 Hybrid	T5 Hybrid + older vacc	1007.5 (956.6-1270.5)	543.5 (464.6-608.2)	100	100
Delta	7	T4 Hybrid	T5 Hybrid + recent vacc	237.4 (169.1-467.4)	311.3 (186-331.9)	100	100

Delta	6	T4 Hybrid + IgN	T5 Hybrid + IgN + older vacc	354.3 (120-782.2)	202.6 (107-429.8)	100	100
Delta	1	T4 Hybrid + IgN	T5 Hybrid + IgN + recent vacc	1656.7 (1656.7-1656.7)	567.9 (567.9-567.9)	100	100
Delta	2	T4 Hybrid + IgN	T5 Hybrid + older vacc	1634.8 (1526.6-1743)	261.5 (195.7-327.3)	100	100
Delta	4	T4 Infected	T5 Hybrid + IgN + recent vacc	22.2 (13.7-41.6)	295.5 (234.9-335.2)	25	100
Delta	21	T4 Infected	T5 Hybrid + recent vacc	22.1 (14.7-41.2)	249 (167.7-367.5)	19	100
Delta	10	T4 Infected	T5 Infected	39.4 (33.8-49.3)	49.9 (42.3-59)	30	50
Delta	79	T4 Infected	T5 Infected + IgN	28.6 (18.4-39.9)	98.5 (67.3-145.3)	13.9	87.3
Delta	3	T4 Infected + IgN	T5 Hybrid + IgN + recent vacc	31.7 (30-43.6)	187.4 (169.9-206.7)	33.3	100
Delta	5	T4 Infected + IgN	T5 Hybrid + recent vacc	49.7 (45.7-59.9)	321.8 (214.4-324.7)	40	100
Delta	7	T4 Infected + IgN	T5 Infected	41.8 (24.2-45.4)	37.7 (25.2-67.8)	14.3	42.9
Delta	47	T4 Infected + IgN	T5 Infected + IgN	42.7 (27.7-60.7)	96.3 (53.5-151.8)	38.3	78.7
Delta	33	T4 recent Vacc	T5 Hybrid + IgN + older vacc	285.1 (158.8-437.6)	475.5 (351.6-983.3)	100	100
Delta	27	T4 recent Vacc	T5 Hybrid + IgN + recent vacc	192.7 (94.7-292.9)	784.9 (459-1233.7)	100	100
Delta	47	T4 recent Vacc	T5 recent Vacc	164.1 (98.3-223)	290.8 (187.6-527.5)	93.6	95.7
Delta	37	T4 recent Vacc	T5 older Vacc	194 (149.2-280.2)	249 (92.1-425.7)	100	91.9
Delta	47	T4 Negative	T5 Hybrid + IgN + recent vacc	0 (0-0)	189.8 (138.9-314.1)	0	85.1
Delta	2	T4 Negative	T5 Hybrid + recent vacc	0 (0-0)	161.3 (140.2-182.4)	0	100
Delta	62	T4 Negative	T5 Infected	0 (0-0)	8.6 (4.9-11.8)	0	0
Delta	148	T4 Negative	T5 Infected + IgN	0 (0-0)	25.9 (11.8-57.1)	0	31.8
Delta	73	T4 Negative	T5 recent Vacc	0 (0-0)	113.1 (56.6-150.1)	0	79.5
Delta	1	T4 Negative	T5 older Vacc	0 (0-0)	35.7 (35.7-35.7)	0	0
Omicron	12	T4 Hybrid	T5 Hybrid + IgN + older vacc	146.2 (104.6-203.9)	144.7 (100.8-217.5)	91.7	100
Omicron	4	T4 Hybrid	T5 Hybrid + IgN + recent vacc	180 (133.1-248.3)	643.9 (559.5-1117.4)	100	100
Omicron	6	T4 Hybrid	T5 Hybrid + older vacc	690.1 (546.5-759.7)	277.4 (236.1-301.2)	100	100
Omicron	7	T4 Hybrid	T5 Hybrid + recent vacc	105 (81.1-155.5)	171 (136.3-233)	100	100
Omicron	6	T4 Hybrid + IgN	T5 Hybrid + IgN + older vacc	156 (41.3-267)	170.8 (125.9-225.2)	66.7	100
Omicron	1	T4 Hybrid + IgN	T5 Hybrid + IgN + recent vacc	913.2 (913.2-913.2)	226.8 (226.8-226.8)	100	100
Omicron	2	T4 Hybrid + IgN	T5 Hybrid + older vacc	1005 (906-1104)	116.9 (102.8-131)	100	100
Omicron	4	T4 Infected	T5 Hybrid + IgN + recent vacc	0 (0-8)	190.9 (154.9-203.9)	0	100
Omicron	21	T4 Infected	T5 Hybrid + recent vacc	0 (0-0)	166.2 (121.7-219.7)	0	100
Omicron	10	T4 Infected	T5 Infected	0 (0-10.8)	22.9 (17.8-30.2)	0	20
Omicron	79	T4 Infected	T5 Infected + IgN	0 (0-4)	88 (61.6-127.1)	0	87.3
Omicron	3	T4 Infected + IgN	T5 Hybrid + IgN + recent vacc	0 (0-15)	151.6 (132.3-152.9)	0	100
Omicron	5	T4 Infected + IgN	T5 Hybrid + recent vacc	5 (0-20)	181.8 (132.3-204.5)	0	100
Omicron	7	T4 Infected + IgN	T5 Infected	0 (0-5.5)	45.7 (26.6-52.7)	0	42.9

Omicron	47	T4 Infected + IgN	T5 Infected + IgN	0 (0-25.5)	95.2 (63.1-154.9)	4.3	80.9
Omicron	33	T4 recent Vacc	T5 Hybrid + IgN + older vacc	108 (73-183.5)	359.2 (239.8-596)	93.9	100
Omicron	27	T4 recent Vacc	T5 Hybrid + IgN + recent vacc	74 (40-116.4)	473.3 (378.7-657.7)	66.7	100
Omicron	47	T4 recent Vacc	T5 recent Vacc	69.1 (36.5-92)	189.8 (105.3-344.2)	68.1	93.6
Omicron	37	T4 recent Vacc	T5 older Vacc	71 (62.8-102)	198.1 (49.4-290.8)	83.8	73
Omicron	47	T4 Negative	T5 Hybrid + IgN + recent vacc	0 (0-0)	178.3 (137.2-253.9)	0	95.7
Omicron	2	T4 Negative	T5 Hybrid + recent vacc	0 (0-0)	82 (74.6-89.4)	0	100
Omicron	62	T4 Negative	T5 Infected	0 (0-0)	30.4 (19.6-52)	0	25.8
Omicron	148	T4 Negative	T5 Infected + IgN	0 (0-0)	81.5 (38.6-160.6)	0	68.2
Omicron	73	T4 Negative	T5 recent Vacc	0 (0-0)	90.2 (54.5-159.1)	0	79.5
Omicron	1	T4 Negative	T5 older Vacc	0 (0-0)	28 (28-28)	0	0

5) **Vaccination status:** It is stated that the “Vaccination status of children and adolescents was either self-reported by children and adolescent on the day of testing in schools or reported by parents/caregivers in online questionnaires». The reliability of self-reported vaccination status in children and adolescents is questionable. Please provide information on the fraction of data coming from self-reporting and parent/caregiver questionnaires, and if possible an assessment of the reliability of self-reporting in the Discussion.

Thank you for highlighting this. We agree and have added this missing information to the manuscript and discussed the validity of reporting in the discussion section (lines 427-433, 575-581)

6) **Estimation of antibody titers by origin:** From what I understand all samples were also tested for anti-N antibodies. Given the type of mRNA-based vaccines available in Switzerland this would enable inferring antibody origin (infection vs. vaccination) by jointly accounting for both antibody types (eg. Zaballa et al. 2023). If feasible I would encourage the authors to add this analysis which would add significantly to the seroprevalence result sections.

All samples were tested for anti-nucleocapsid IgG. This is mentioned in the methods section, we have accounted for both anti-spike IgG and anti-nucleocapsid IgG when categorising children and adolescents by exposure status (lines 130-137, 587-595): “To assess the evolution of anti-spike IgG and neutralising antibody titres, we categorized children and adolescents from the longitudinal cohort into four groups according to their vaccination and infection status. Children and adolescent never testing positive for anti-spike IgG were categorised as seronegative, unvaccinated children and adolescent ever testing positive for anti-spike IgG as infected, vaccinated children and adolescent testing negative for anti-spike IgG prior to vaccination and never testing positive for anti-nucleocapsid IgG as vaccinated, and children and adolescents testing seropositive before getting vaccinated, or were vaccinated and tested positive for anti-nucleocapsid-IgG antibodies as hybrid.”

7) **Estimation of binding antibody decay times:** The computation of the antibody half-life is based on a subsample of participants of the longitudinal cohort which consists of infected-only participants with no history posterior history of infection as quantified by anti-N serology. Could the authors comment whether this could introduce a form of selection bias for participants with particular immune responses which could affect the representativeness of the estimate? Regarding the sensitivity analysis, **were the two different subsets of participants comparable in terms of age/sex?** Another point regards the selection of the longitudinal cohort for the decay time analysis. Were participants with three time points, but no data at T5 removed from the analysis, as the description in the methods seems to imply? If so, please give a rationale for this, and clarify the specific set of participant retained by providing data on the number of participants with different number of repeated samples.

We agree that children and adolescents with particular immune responses can always affect the representativeness of the estimate. However, our sample comprises predominantly healthy school children and adolescents, and our data suggests a homogenous antibody decay in both primary and sensitivity analyses (Supplementary Figures 4a-d). We adapted the manuscript accordingly (lines 343-349).

You are correct that we excluded children and adolescents without T5 measurements as one main goal of the paper was to focus on the Omicron period. Based on your comment, we built an additional cohort (sensitivity cohort) as sensitivity analysis that was based on children and adolescents that participated in four or more testing rounds regardless of whether they participated in T5. Compared to the main cohort, half-life estimates were very similar for the longer and shorter time window. We also found that the sex distribution and median age in both cohorts, each with a short and a longer time window were similar (range 42-45% males, and median age 11, IQR 9-12 and 9-14, respectively, for the shorter and longer time window.) We are therefore confident that our findings are robust and do not seem to be biased by non-comparable groups among the different cohorts and time windows. The corresponding figures have now been included in the Supplementary material (Supplementary Figure 4a-d) and the findings are discussed in the discussion (lines 174-195, 321-349, 632-650).

8) **Model equations and code sharing:** For clarity, please add all model equations used in the study in an appendix. I would also urge authors to add a public repository with analysis code used in this study as well as minimal datasets to illustrate their use.

We have now added all the model equations in the supplementary method. Further we have also submitted our analysis code and the corresponding dataset. Please see the documents submitted (SourceData & Supplementary Code).

9) **Participation rates:** Please provide individual-level participation rates in the study.

We have added the participation rates per testing round in the result section (lines 117-118) and also in the Supplementary Table 1.

Other points of importance:

10) **Title:** The word “immunity” suggests that the study address protection, which this analysis does not answer in its current form. I would therefore suggest to replace with something that indicates detection through measurements.

We have adapted the title as suggested.

11) **Omicron sub-variants:** Given the emergence of Omicron subvariants and their distinct characteristics, systematically mentioning what subvariants were analyzed in this study would better enable to contextualize the study in the existing literature.

We used the Spike trimer protein from the Omicron BA.1 sub-variant, which we have now added to the manuscript (lines 536-537). In our vast experience with the laboratory at the University Hospital of Lausanne, where they tested more than 250'000 samples, there was not a significant difference between the neutralizing antibody response between the Omicron BA.1 and BA.2 variant and only a small relative decrease in neutralizing capacity against the Omicron BA.4 variant. The largest differences that we and other have observed was found between the pre- (i.e., 2019-nCoV, Alpha, Beta, Delta) and post-Omicron variants.

12) **Figure 3:** It is very hard for me to make sense of the longitudinal trajectories with these visualization, which is the main aim of the figure from what I understand. I would suggest following common practice in representing longitudinal antibody trajectories using dots and lines per participant, with the x axis being the time of sample collection (eg. Figure 1 in Iyer et al. 2020).

Thank you for the suggestion. We agree and have added a Figure 1b with the longitudinal antibody trajectories to the main body of the paper as suggested.

13) **Figure 4/5:** It is unclear to me whether T4 Infected/T5 Infected are participants with infections before T4 AND occurring between T4 and T5, or whether it is not possible to differentiate this here. Please add this information to the caption/discussion.

T4 infected denotes individuals that tested seropositive anti-spike IgG antibodies (and were unvaccinated) prior to the T4 assessment, T5 infected denotes individuals that tested seropositive (and were unvaccinated) due to anti-spike IgG antibodies between T4 and T5. We now clarified this difference in the captions (Figure 2 (line 262-269) and 3 (line 297-305)). Further, we also discuss reinfection between T4 and T5 among infected and vaccinated children and adolescents (see Supplementary Table 8a-b) and discussion (lines 317-318, 398-411).

14) **Results in Supplementary table 6:** It is unclear for me how to read the table. In particular, how can the column anti-N negative have numbers for percentage increase/decrease? And how was the 25% threshold determined? I believe that this is an

important part of the results, but would require more explanation to value it in the main text. (Also note the typo in the caption referring to “older infection”).

We agree that the table was difficult to read, we apologize. We hope that you find the new format more understandable. This table is meant to describe re-infections between T4 and T5 for infected (i.e., vaccine naïve) children and adolescents, those with older (prior to T4) and recent (between T4 and T5) vaccination, irrespective of infection prior to T4. The occurrence of reinfection was based on an a priori defined, but arbitrary cut-off of 25% or higher increase in anti-spike IgG titers, and/or the existence of anti-nucleocapsid IgG antibodies in the infected group and those with older vaccination (prior to T4). For participants with recent vaccination, only the criterium of the existence of anti-nucleocapsid IgG was required as the increase in anti-spike IgG titers was likely related to the additional vaccination. The cut-off of $\geq 25\%$ was judged as relevant increase consistent with a re-infection (in the absence of vaccination) between two testing points using data from population-based cohorts throughout the pandemic and by considering any potential measurement error. In order to test the robustness of this cut-off, we added some sensitivity analyses with variable cut-offs of $\geq 15\%$ and $\geq 35\%$ increase. This information is now included in the manuscript in the result and method section (lines 232-255, 398-411, 625-630, Supplementary Figure 8a-b).

15) Comparison of neutralization capacity results with Zaballa et al. 2023: In Zaballa et al. 2023, much lower neutralization capacity among school-aged children were found using the same neutralizing test, in particular below 12 years of age, than in this study. Could the authors comment on the potential reasons these differences may exist, and their implications for the interpretation of this study’s results? I encourage the authors to provide more details (equations, code, etc) for future comparison of results between these two studies.

Thank you for raising this point. We have now added a discussion regarding the differences between our findings and those reported by Zaballa et al ³. The study by Zaballa et al found that neutralising activity, especially for Omicron BA.1, was much lower in children and adolescents compared to our results despite the use of the same assay ⁴. It is possible that neutralisation in our children and adolescents with vaccination and hybrid immunity was higher due to undetected or repeated infections with Omicron which has been shown to boost neutralization ⁵. Yet, any differences between their and our study may also have resulted from systematic differences in study populations, the technical setup or random chance. As this is a very important point, we also extended the discussion section in this respect (lines 351-379).

As also suggested, we have now added all equations to the supplementary material and submitted the code and data in separate files (SourceData & Supplementary Code).

16) Tone of results/discussion/conclusion: I invite the authors to tone down the novelty and importance of results throughout the text as these are not warranted in my opinion, and to leave more space for the reader’s interpretation and evaluation.

We agree and toned down novelty and importance of the study in results, the discussion and conclusion section.

In order of appearance:

- L. 66-68: the message conveyed by this sentence is not clear to me

We agree and have adapted the sentence accordingly. (lines 60-62)

- L. 71: 5 hospitalizations (even short) – or 3 as stated line 119? - in a sample of healthy children and adolescents seems high, and the word “only” seems inappropriate to me in this statement. Please provide contextual data on hospitalization rates in children/adolescents for SARS-CoV-2 infections in Switzerland or internationally.

Based on your suggestion, we added further details on the hospitalised children and adolescents in the Supplementary Table 3 as well as contextual data on international hospitalisation rates (lines 124-126, 372-379, 582-585).

- L. 86-87 (ref 1-6): Multiple seroprevalence studies were done in Switzerland, including the Omicron period which provide relevant context to this study (eg. Zaballa et al. 2023). If the authors chose to exclude these studies from the reference list please provide a rational for doing so.

We added these studies to our references (lines 80-92).

- L. 103-105: Please add a reference to this statement.

We have added a reference as suggested (lines 101-104).

- L. 107: the phrase “longitudinal development” is vague given that it could refer to the longitudinal cohort only. I would recommend keeping the word longitudinal for serial participant-level measurements and using “repeated cross-sectional” for the time evolution of seroprevalence.

We agree and now use the terms “longitudinal” and “repeated cross-sectional” accordingly (lines 106-111).

- L. 114: “between 1876 and 2500” please check consistency with Figure 1/Table 1 which say 1875.

Thank you for the comment. We checked all the numbers again throughout the whole manuscript.

- Figure 2. Line “unvaccinated” in red and not light blue as indicated in legend.

Thank you for highlighting this. We adapted it as suggested (Supplementary Figure 2).

- L. 163-169: In these results of antibody detection in time, how were possible re-infections accounted for? Credible intervals should be provided, in particular for the last two groups which had very limited sample sizes.

Here we did not differentiate whether there was a first infection, reinfection, vaccination, or a combination, as we wanted to provide an overall picture. However, we state that the increase could be due to (re-)infection, vaccination, or a combination of both (lines 164-166). We added the confidence intervals for the provided numbers (lines 156-161).

- L. 171-173: Please provide quantitative support for these statements.

We added quantitative support for these statements (lines 166-172).

- L. 174: It seems misleading to speak of “duration of protection” here, although anti-S IgG have indeed been shown to be correlates of protection in some cases. Please nuance this statement.

Indeed, this formulation is somewhat misleading and was therefore adjusted (line 174).

- L. 202: Cf. Major points. How can vaccinated-only participants increase in both binding and neutralizing antibody levels between T4 and T5? Please develop in the discussion.

Thank you for the comment. We agree and have provided additional information to the results and the discussion (lines 225-228, 282-285, 363-379, see comment number 4 for more details). As shown in the figures (Figure 1 and 2; and Tables 1 and 2), we were not able to detect a difference in anti-spike IgG titres nor in neutralising response in children and adolescents with a recent vaccination compared to those with an older vaccination. This may be due to the misclassification of children and adolescents from the hybrid into the vaccinated group, since anti-nucleocapsid IgG antibodies wane quickly and the response is weaker when vaccinated. We reported this miss-classification throughout the manuscript. This fact is also taken up in the limitation section (lines 423-438).

- L. 211-214: This seems to be an important point. Cf. Comment in other important points on Table S6. To dig this further, could a sub-analysis also be done using binding/neutralization capacity as exposure variable?

We agree and have provided additional information on reinfection rates throughout the manuscript and used different thresholds (see comment number 14). To detect and quantify infections and reinfections among children and adolescents between T4 and T5, we used an a priori but arbitrary threshold of 25% or higher increase in anti-spike IgG titres. We judged this threshold to reflect a relevant increase consistent with an infection (in the absence of vaccination), based on observations from several population-based cohorts over the course of the pandemic, and by considering any possible measurement error. We then conducted sensitivity analyses using a 15% and 35% threshold. As already described in the previous version of the manuscript re-infection rates were higher among the infected (i.e., vaccine-naïve) than the vaccinated children and adolescents. However, the clinical relevance of this finding is not clear and is now discussed in the discussion section (lines 232-255, 398-411, 625-630, Supplementary Figure 8a-b). After reviewing the data on reinfections and considering the subgroups (infected, recent vaccinated and older vaccinated children and adolescents), we believe that there is no need to conduct the suggested sub-analysis.

- L. 272-272: Given the number of studies on school-aged seroprevalence since the Omicron subvariants I think that “remarkable” is somewhat of an overstatement.

We adapted this accordingly.

- L. 276-278: “These findings are consistent with other international studies in children and adolescents also reporting high seroprevalence and titre levels by mid 2022”. It seems relevant to include a comparison to results from Switzerland here (Zaballa et al. 203). As noted above, please provide a comparison of results and discussion of differences.

As mentioned in comment number 15, we have added a comparison of our results to those by Zaballa et al in the discussion section (lines 351-379).

- L. 292: See Dowel et al. for a comprehensive analysis of anti-SARS-CoV-2 immune response in children.

We added a comparison of our findings regarding antibody decay with those reported by Dowel et al to the discussion section (lines 389-396).

- L. 320: Rates of long-COVID have been reported to be an order of magnitude larger (eg. Dumont et al. 2022), which may be worth mentioning as a public health relevant outcome.

We are not referring to Long COVID in this section, but rather on the severe course of an acute SARS-CoV-2 infection in children and adolescents and reason about the meaning of high antibody titres potentially preventing a severe course, as reported in adults. We have now adjusted this section to make it clearer what exactly we are referring to (lines 372-379). Based on our imprecise data regarding vaccination, timing of infections and symptomatology, we cannot report on Long Covid based on history of infections, vaccination, or both.

- L. 322-329: All preventive measures were lifted in February/March 2022, and seroprevalence estimates almost reached 100% in July 2022. It is questionable whether this can be considered as a public health success, in particular regarding the risks of long COVID, and the potential longer-term risks of infection. I invite the authors to tone down or remove this paragraph as I do not believe it is suited for a scientific publication in an academic journal.

We agree that this section is somewhat strongly formulated. We were in close contact with the Federal Office of Public Health (FOPH) Switzerland, kept them up to date and informed them regarding our results. As we do not know what decisions were taken based on our data or other considerations, we dropped the respective sentences.

- L. 335: Was there any information in this study to quantify asymptomatic rates?

Given the inconsistent reporting of symptoms, confounding with symptoms by other infections and vaccination and limited data on diagnosis of SARS-CoV-2 infections, it was not possible to quantify the asymptomatic rates in children and adolescents in our study. We added this to our limitation section (lines 447-449).

- L. 337: I would invite the authors to avoid “first study”-type sentences of this type which do not contribute to the scientific value of the paper.

We adapted as suggested. “First study” does not appear anymore in the whole manuscript.

- L. 392: Please state from which age masks were mandatory. I believe that young children never had to wear a mask at school.

In the canton of Zurich⁶, masks were mandatory for all school children and adolescents from December 2021 to mid-February 2022 during the first peak of the Omicron wave. We now adapted the sentence accordingly (lines 482-484).

- L. 414: Cf. Point about the definition of the longitudinal cohort with respect to the computation of antibody decay time

We are not entirely sure what the reviewer is referring to. If this is what the reviewer refers to, we have an extra paragraph in the method section, where we describe the children and adolescents selected for the antibody decay estimation (lines 632-650). We also added a sensitivity cohort that was not requiring the T5 measure as criterium and ran a sensitivity analysis (see comment number 7). We also report on differences in sex and age among the different cohorts, each covering a shorter and longer time window. In summary, decay times were similar for the two cohorts and each for the shorter and longer time window between cohorts, and characteristics of included children and adolescents among cohorts and time windows did not differ for sex and age distribution (lines 174-195, 321-349).

- L. 427: Are any studies available on decay of sensitivity with time?

The laboratory at the University Hospital of Lausanne that was performing all our analyses has been using this test for 3 years now and in addition regularly performs quality controls with references sera to ensure the stability of the test over time. Based on the literature, performance of the assays used in our study is stable and shows a minor decay up to 8 months after infection². Whether decay time remain minimal over longer time is not known.

We therefore added a comment that we did not control for decay of sensitivity with time as this decay seems to be minimal over months². (lines 531-534)

- L. 439: Did questionnaires ask for COVID-19-related hospitalizations and symptoms? If not, how was the information about hospitalization reported in the abstract retrieved?

Hospitalizations and symptoms were included in the questionnaires. However, we did not report symptoms in children and adolescents since this information was limited by inconsistent reporting, confounding with other infections and vaccination side effects, and missing data in the questionnaires. Regarding hospitalizations, participants were asked to report any hospitalizations that were related to episodes of symptoms potentially, but not exclusively related to a SARS-CoV-2 infection and which were unrelated to chronic diseases or known allergies. Collected data regarding hospitalization was further clarified by contacting parents or caregivers by phone.

We have now adapted the method section accordingly and added a detailed table of the children and adolescents hospitalised (Supplementary Table 3, lines 582-585).

- L. 467: Was household information available? If so, what was the rationale for not accounting for household clustering? Also, please report posteriors for all inferred model parameters of importance in the supplement. Additionally, please indicate how convergence was assessed.

We did not collect household information and therefore are unable to adjust for household clustering which was, based on the average number of children per Swiss family (1.5) and our random selection of children within schools and school classes not a big issue.

We have added all models used to the Supplementary Method section.

- Figure 5: In panel (a) it seems like all values for T4 are capped at an upper limit. Does the neutralizing test have an upper limit of quantification? If so, please mention in the methods.

The values are capped at an upper limit of 2430 due to the dilution series in the laboratory generating non-significant values. We agree that this may have led to misinterpretations in Figure 3. We now consistently adapted all the values in the figures and tables accordingly and mentioned this limitation in the methods section (lines 568-570).

- Supplementary Figure 3/Table 5: Interpretation of Roche-S results is not straightforward to the scientific community at large. I would suggest to transform these results to International Units instead.

We have adapted the method section accordingly and provided all tables and figures in the requested WHO units per millilitre (U/ml for Roche Elecsys anti-spike IgG) an international unit (see Supplementary Figures and Tables, lines 605-609).

References

Doucette, E.J., Gray, J., Fonseca, K., Charlton, C., Kanji, J.N., Tipples, G., Kuhn, S., Dunn, J., Sayers, P., Symonds, N. and Wu, G., 2023. A longitudinal seroepidemiology study to evaluate antibody response to SARS-CoV-2 virus infection and vaccination in children in Calgary, Canada from July 2020 to April 2022: Alberta COVID-19 Childhood Cohort (AB3C) Study. *Plos one*, 18(4), p.e0284046.

Dumont, R., Richard, V., Lorthe, E., Loizeau, A., Pennacchio, F., Zaballa, M.E., Baysson, H., Nehme, M., Perrin, A., L'Huillier, A.G. and Kaiser, L., 2022. A population-based serological study of post-COVID syndrome prevalence and risk factors in children and adolescents. *Nature communications*, 13(1),

Dowell, A.C., Butler, M.S., Jinks, E., Tut, G., Lancaster, T., Sylla, P., Begum, J., Bruton, R., Pearce, H., Verma, K. and Logan, N., 2022. Children develop robust and sustained cross-reactive spike-specific immune responses to SARS-CoV-2 infection. *Nature immunology*, 23(1), pp.40-49.

Frei, A., Kaufmann, M., Amati, R., Butty Dettwiler, A., von Wyl, V., Annoni, A.M., Pellaton, C., Pantaleo, G., Fehr, J.S., D'Acremont, V. and Bochud, M., 2022. Development of hybrid immunity during a period of high incidence of infections with Omicron subvariants: A prospective population based multi-region cohort study. *medRxiv*, pp.2022-10.

Han, M.S., Um, J., Lee, E.J., Kim, K.M., Chang, S.H., Lee, H., Kim, Y.K., Choi, Y.Y., Cho, E.Y., Kim, D.H. and Choi, J.H., 2022. Antibody responses to SARS-CoV-2 in children with COVID-19. *Journal of the Pediatric Infectious Diseases Society*, 11(6), pp.267-273.

Iyer, A.S., Jones, F.K., Nodoushani, A., Kelly, M., Becker, M., Slater, D., Mills, R., Teng, E., Kamruzzaman, M., Garcia-Beltran, W.F. and Astudillo, M., 2020. Persistence and decay of human antibody responses to the receptor binding domain of SARS-CoV-2 spike protein in COVID-19 patients. *Science immunology*, 5(52), p.eabe0367.

Zaballa, M.E., Perez-Saez, J., de Mestral, C., Pullen, N., Lamour, J., Turelli, P., Raclot, C., Baysson, H., Pennacchio, F., Villers, J. and Duc, J., 2023. Seroprevalence of anti-SARS-CoV-2 antibodies and cross-variant neutralization capacity after the Omicron BA. 2 wave in Geneva, Switzerland: a population-based study. *The Lancet Regional Health—Europe*, 24.

References

1. Bürzle, O. *et al.* Adverse effects, perceptions and attitudes related to BNT162b2, mRNA-1273 or JNJ-78436735 SARS-CoV-2 vaccines: Population-based cohort. *NPJ Vaccines* **8**, 61 (2023).
2. Perez-Saez, J. *et al.* Persistence of anti-SARS-CoV-2 antibodies: immunoassay heterogeneity and implications for serosurveillance. *Clinical Microbiology and Infection* **27**, 1695.e7-1695.e12 (2021).
3. Zaballa, M.-E. *et al.* Seroprevalence of anti-SARS-CoV-2 antibodies and cross-variant neutralization capacity after the Omicron BA.2 wave in Geneva, Switzerland: a population-based study. *The Lancet Regional Health - Europe* **24**, 100547 (2023).
4. Fenwick, C. *et al.* A high-throughput cell- and virus-free assay shows reduced neutralization of SARS-CoV-2 variants by COVID-19 convalescent plasma. *Sci Transl Med* **13**, (2021).
5. Dowell, A. C. *et al.* Immunological imprinting of humoral immunity to SARS-CoV-2 in children. *Nat Commun* **14**, 3845 (2023).
6. Kanton of Zurich. Erweiterte Maskentragpflicht an den Schulen. *Canton of Zurich* <https://www.zh.ch/de/news-uebersicht/medienmitteilungen/2021/12/erweiterte-maskentragpflicht-an-den-schulen.html> (2021).

REVIEWERS' COMMENTS

Reviewer #1 (Remarks to the Author):

I think the authors provided enough responses to both reviewers

Reviewer #2 (Remarks to the Author):

I thank the authors for the consideration of the previous round of comments, for the additional information provided in the rebuttal, and for their efforts in incorporating changes in the manuscript. I believe it has significantly improved the presentation and value of the work.

Based on the authors' reply, I still had a two outstanding main comments and two minor ones.

Main points:

- Categorization of participants and main results: I think that the more indepth discussion of this important limitation will enable readers to better grasp the relevance and limitations of the study, and thank the authors for including it. I would however clearly state the potential effect of this miss-classification (ie. over-estimation of "vaccination only" immune responses) in the Discussion (eg. l. 438), and would nuance all statements in the text that directly compare the two classes of hybrid and vaccination as equivalent (eg. l. 314).
- Omicron variant specification: I understand the reply of the authors regarding neutralization capacity, I however insists on specifying the subvariant throughout the text, captions and supplementary material, possibility mentioning the point on neutralization capacity for other subvariants the authors refer to in the discussion supported by references.

Other points:

- Reply to point (5): could the authors specify in the manuscript how vaccination validation was done?
- Reply to point (13): Could you please expand on the reasons for the differences, random chance seems unlikely, quantify statistically? What do you mean with technical setup? why should this affect only children? study populations are similar in terms of age classes, what other population-level factors could explain the difference? What about time to infection? I understand these details may not be necessary to include in the manuscript, but I would like to see a more indepth discussion in the rebuttal, especially if the review material will be published alongside the manuscript.

REVIEWERS' COMMENTS

Reviewer #1 (Remarks to the Author):

I think the authors provided enough responses to both reviewers

We thank the reviewer once again for their valuable time and their insightful comments.

Reviewer #2 (Remarks to the Author):

I thank the authors for the consideration of the previous round of comments, for the additional information provided in the rebuttal, and for their efforts in incorporating changes in the manuscript. I believe it has significantly improved the presentation and value of the work.

Based on the authors' reply, I still had two outstanding main comments and two minor ones.

Main points:

1) Categorization of participants and main results: I think that the more indepth discussion of this important limitation will enable readers to better grasp the relevance and limitations of the study, and thank the authors for including it. I would however clearly state the potential effect of this miss-classification (ie. over-estimation of "vaccination only" immune responses) in the Discussion (eg. l. 438), and would nuance all statements in the text that directly compare the two classes of hybrid and vaccination as equivalent (eg. l. 314).

Thank you for your valuable comments and time to review our manuscript.

Your point is well taken. Let us summarize and comment:

In the Results section we describe our findings (e.g., similar titre levels) among the hybrid and vaccinated groups (e.g., starting from line 201).

In the first paragraph of the Discussion (starting from line 274) you find a summary of these findings, not intended to be discussed here.

Further down (starting from line 325) we discuss our findings compared to the literature. We specifically highlighted that undetected or repeated infections may have led to misclassification of the vaccinated only group (starting from line 328) possibly leading to false high anti-spike IgG titres and neutralisation and thus an overestimation of immune responses in the "vaccine only" group.

Based on your comment, we adapted the discussion and limitations section accordingly (starting from line 403). We feel that these changes do sufficiently address your requests.

2) Omicron variant specification: I understand the reply of the authors regarding neutralization capacity, I however insists on specifying the subvariant throughout the text,

captions and supplementary material, possibility mentioning the point on neutralization capacity for other subvariants the authors refer to in the discussion supported by references.

We gladly adapted the manuscript defining the subvariant of Omicron and providing the information that subvariants may elicit different immune responses (starting from line 317).

Other points:

3) Reply to point (5): could the authors specify in the manuscript how vaccination validation was done?

We adapted this as suggested (starting from line 393).

4) Reply to point (13): Could you please expand on the reasons for the differences, random chance seems unlikely, quantify statistically? What do you mean with technical setup? why should this affect only children? study populations are similar in terms of age classes, what other population-level factors could explain the difference? What about time to infection? I understand these details may not be necessary to include in the manuscript, but I would like to see a more indepth discussion in the rebuttal, especially if the review material will be published alongside the manuscript.

We added a more in-depth discussion on the comparison of our study to the Zaballa et al ¹ study here. As reported also in the main article, we do agree that there appears to be a lower neutralizing capacity in the population reported by Zaballa et al. compared to our cohort. We still feel strong about our previous statement: Although the same assay was used for these two studies, the reagents, instruments, and operators were different, each of which could have contributed to the lower IC50 values observed by Zaballa et al. This conclusion is based on different factors that are described in the following section.

The Spike-ACE2 surrogate neutralization assay was developed at the Cantonal Hospital of Vaud (CHUV), and all of the Luminex testing to establish the cut-off limits for neutralizing activity in Fenwick et al ² were performed at the CHUV) This assay was transferred to the CHUV diagnostics department for routine testing of patient samples and these same hospital technicians evaluated the serum samples from the school-aged children in our study. Briefly, all of the reagents used for these assays were stringently tested and validated before use. The validation studies included testing of positive control reference serum samples and purified anti-SARS-CoV-2 neutralizing monoclonal antibodies for new Spike coupled beads, where neutralizing IC50 values were compared to the expected potencies observed in Spike pseudo-typed lentiviral neutralization assays. In addition, a concentration response of anti-SARS-CoV-2 neutralizing monoclonal antibodies was generated on each 96-well plate used for serum sample evaluation along with a negative control consisting of a pool of healthy donor serum samples collected prior to the 2019 pandemic.

Despite this stringent methodology it appears that all neutralizing antibody IC50 values for adults and children alike were lower and closer to the serum dilution IC50 cut-off of 50 established by Fenwick et al ². In the comparative analysis of mean neutralizing activities, there would not be a difference in the interpretation of results. However, in younger children

with lower levels of neutralizing antibodies, there would be a higher percentage of individuals below this cut-off of 50. Our analysis of serum neutralizing antibody activities in Figure 3 was comparative across the different groups (Hybrid, Vaccinated and Infected) and evaluated times so there will not be any effect on the interpretation of our results.

Furthermore, it is possible that neutralisation in our children and adolescents with vaccination and hybrid immunity was higher due to undetected or repeated infections with Omicron which has been shown to boost neutralization ³.

Zaballa et al. recruited adults from a randomly selected sample that had previously taken part in one of their serosurveys. Subsequently, they extended invitations to their children and adolescents to participate in the study and participation rate was 12.5%. Neutralisation was measured only in a subset of participants. This approach potentially introduced a selection bias, possibly contributing to differences in results between our study and theirs. Additionally, this difference in findings could also be due to chance since the sample size of the study by Zaballa et al was substantially lower than ours.

Finally, due to the study's reliance on serology testing in the absence of PCR test data in both cohorts we were unable to determine the exact timing of infection among our participants. Thus, different time periods since infection among the two studies could have contributed to different levels of antibodies and neutralisation. However, the implications of these divergences remain uncertain. We added this missing information to the main manuscript (starting from line 319).

Remark on the not answered parts of this comment:

“Why should this affect only children?”: We cannot comment on that, since we do not have adults in our study.

References:

1. Zaballa, M.-E. *et al.* Seroprevalence of anti-SARS-CoV-2 antibodies and cross-variant neutralization capacity after the Omicron BA.2 wave in Geneva, Switzerland: a population-based study. *The Lancet Regional Health - Europe* **24**, 100547 (2023).
2. Fenwick, C. *et al.* A high-throughput cell- and virus-free assay shows reduced neutralization of SARS-CoV-2 variants by COVID-19 convalescent plasma. *Sci Transl Med* **13**, (2021).
3. Dowell, A. C. *et al.* Immunological imprinting of humoral immunity to SARS-CoV-2 in children. *Nat Commun* **14**, 3845 (2023).